

# Aerosol and physical atmosphere model parameters are both important sources of uncertainty in aerosol ERF

Leighton Regayre[1], Jill Johnson[1], Masaru Yoshioka[1], Kirsty Pringle[1], David Sexton[2], Ben Booth[2], Lindsay Lee[1], Nicolas Bellouin[3], and Kenneth Carslaw[1]

[1]Institute for Climate and Atmospheric Science, School of Earth and Environment, University of Leeds, Leeds, LS2 9JT, UK.
[2]UK Hadley Centre Met Office, Exeter, Fitzroy Road, Exeter, Devon, EX1 3PB, UK.
[3]Department of Meteorology, School of Mathematical & Physical Sciences, Faculty of Science, University of Reading, Reading, RG6 6BB, UK.

*Correspondence to:* Leighton Regayre (L.A.Regayre@leeds.ac.uk)

**Abstract.** Changes in aerosols cause a change in net top-of-the-atmosphere (ToA) short-wave and long-wave radiative fluxes,
rapid adjustments in clouds, water vapour and temperature, and cause an effective radiative forcing (ERF) of the planetary
energy budget. The diverse sources of model uncertainty and the computational cost of running climate models make it diffi-
cult to isolate the main causes of aerosol ERF uncertainty and to understand how observations can be used to constrain it. We
explore the aerosol ERF uncertainty by using fast model emulators to generate a very large set of aerosol-climate model vari-
ants that span the model uncertainty due to twenty-seven parameters related to atmospheric and aerosol processes. Sensitivity
analyses shows that the uncertainty in the ToA flux is dominated (around 80%) by uncertainties in the physical atmosphere
model, particularly parameters that affect cloud reflectivity. However, uncertainty in the change in ToA flux caused by aerosol
emissions over the industrial period (the aerosol ERF) is controlled by a combination of uncertainties in aerosol (around 60%)
and physical atmosphere (around 40%) parameters. Four of the atmospheric and aerosol parameters that cause uncertainty in
short-wave ToA flux (mostly parameters that directly scale cloud reflectivity, cloud water content or cloud droplet concentra-
tions) also account for around 60% of the aerosol ERF uncertainty. The common causes of uncertainty mean that constraining
the modelled planetary brightness to tightly match satellite observations changes the lower 95% credible aerosol ERF value
from -2.65 W m$^{-2}$ to -2.37 W m$^{-2}$. This suggests the strongest forcings (below around -2.4 W m$^{-2}$) are inconsistent with ob-
servations. These results show that, regardless of the fact that the ToA flux is an order of magnitude larger than the aerosol ERF,
the observed flux can constrain the uncertainty in ERF because their values are connected by constrainable process parameters.
The key to reducing the aerosol ERF uncertainty further will be to identify observations that can additionally constrain indi-
vidual parameter ranges and/or combined parameter effects, which can be achieved through sensitivity analysis of perturbed
parameter ensembles.

## 20   1   Introduction

Large aerosol radiative forcing uncertainty has persisted through all Intergovernmental Panel on Climate Change assessment
reports since 1996 despite substantial developments in climate model complexity (Flato et al. 2013, Section 9.1.3), numerous



intercomparison projects (Randles et al., 2013; Tsigaridis et al., 2014; Kim et al., 2014; Mann et al., 2014; Pan et al., 2015;
Lacagnina et al., 2015; Kipling et al., 2016; Ghan et al., 2016; Koffi et al., 2016), and enormous investments in observing
systems (Khain et al., 2000; Lacagnina et al., 2015; Seinfeld et al., 2016; Reddington et al., 2017). Reducing aerosol forcing
uncertainty has therefore proven to be one of the most challenging and persistent problems in atmospheric science.
Reduction of uncertainty in aerosol effective radiative forcing (ERF) is an important objective, not least because it would
improve climate change projections (Andreae et al., 2005; Myhre et al., 2013; Collins et al., 2013; Tett et al., 2013; Seinfeld
et al., 2016). An improved understanding of the causes of uncertainty would also help to prioritise model developments, sug-
gest fruitful analyses across multiple models, and point to potential new observations to constrain the uncertainties. However,
the task remains challenging for multiple reasons. For example, aerosol ERF is usually quantified with reference to a period
pre-dating the satellite era (usually 1850 or 1750) meaning it is not a directly observable quantity. Satellite-derived observations
of present-day aerosol-cloud relationships have the potential to constrain the aerosol ERF uncertainty, but require an improved
understanding of aerosol changes over the industrial period (Gryspeerdt et al., 2017). Some of the ERF uncertainty might
therefore be irreducible unless pristine present-day environments are shown to be a good proxy for pre-industrial conditions
(Carslaw et al., 2013; Hamilton et al., 2014; Carslaw et al., 2017). Furthermore, aerosol ERF depends on many poorly under-
stood interactions of aerosols with components of the physical climate system. Important sources of uncertainty are known
to be aerosol emission fluxes (Granier et al., 2011), representations of complex sub-grid processes such as clouds (Haerter
et al., 2009; Lohmann and Ferrachat, 2010; Guo et al., 2013; Gettleman et al., 2013; Golaz et al., 2013; Neubauer et al., 2014;
Lohmann, 2017), precipitation responses (Tost et al., 2010; Croft et al., 2012; Michibata and Takemura, 2015), aerosol pro-
cesses (Croft et al., 2012; Textor et al., 2006, 2007; Storelvmo et al., 2009; Kasoar et al., 2016), radiation calculations (Stier
et al., 2013; Wilcox et al., 2015) and subsequent feedbacks on atmospheric dynamics (Booth et al., 2012; Bollasina et al., 2013;
Kirtman et al., 2013; Villarini and Vecchi, 2013; Allen et al., 2014) and surface temperatures (Golaz et al., 2013).
Our intention here is to constrain aerosol ERF uncertainty by pursuing a 'bottom-up' approach that explores the underlying
process uncertainty. This approach provides a set of observationally plausible model variants with which near-term climate
simulations could be performed. Although a lower limit to the global mean aerosol ERF might be found using a 'top-down'
approach and historical temperature trends (Stevens, 2015), inferences made about the climate system are very sensitive to the
simplifying assumptions that are made in top-down approaches (Knutti et al., 2008; Kretzschmar et al., 2017). More impor-
tantly, such methods do not provide a model with which to make improved climate projections and they provide no information
about regional variations in forcing, which are known to be important drivers of climate variability (Chalmers et al., 2012; Dun-
stone et al., 2013; Shindell et al., 2013; Kirtman et al., 2013; Bollasina et al., 2013). Therefore, bottom-up methods that quantify
aerosol ERF using global climate models whose performance and uncertainty are constrained by observations are required.

Multi-model studies (or model intercomparison projects, MIPs) can provide some information about ERF uncertainty be-
cause a set of models with different dynamical cores and physical process parametrisations produces a range of aerosol re-



sponses. However, such opportunistic sampling has three main disadvantages. Firstly, inter-model comparisons often include models with vastly different degrees of complexity (Collins et al., 2013). For example, aerosol indirect effects are not represented in many of the models included in such studies and this artificially inflates multi-model forcing uncertainty (Bellucci et al., 2017). Secondly, multiple members of an inter-model comparison will share key modules and behaviours (Pennell and Reichler, 2010; Collins et al., 2010; Knutti et al., 2013). This leads to compensating effects between groups of models with shared structural errors that causes the multi-model mean to outperform the majority of individual models across a range of climate metrics (Rougier, 2016). Thirdly, a small set of models (perhaps around twenty) cannot possibly sample the effects of dozens of interacting uncertain processes in the individual models (Carslaw et al., 2018). Therefore, inter-model comparisons do not provide statistically representative samples (Sexton et al., 2012; Knutti et al., 2013; Collins et al., 2013), making it difficult to draw inferences about the causes of aerosol ERF uncertainty and the robustness of any observational constraint. Leading experts subjectively assess the uncertainty in aerosol forcing as being larger than that quantified by multi-model studies (Morgan et al., 2006).

A complementary approach to exploring aerosol ERF uncertainty in multiple models is to systematically explore the uncertain in underlying parameters and processes within a single model. Much progress has been made in understanding the causes of uncertainty in state variables related to aerosol ERF, such as cloud-active aerosol concentrations (Lee et al., 2011, 2012, 2013; Samset et al., 2014; Mann et al., 2014; Shrivastava et al., 2016; Kipling et al., 2016), precipitation (Lebo and Feingold, 2014; Qian et al., 2015; Johnson et al., 2015) and ToA radiative fluxes (Shiogama et al., 2012; Zhou et al., 2013; Randles et al., 2013). Furthermore, important sources of aerosol forcing uncertainty (in the absence of rapid atmospheric adjustments) have been identified (Schulz et al., 2006; Haerter et al., 2009; Lohmann and Ferrachat, 2010; Carslaw et al., 2013; Myhre et al., 2013; Regayre et al., 2014, 2015). However, no study has comprehensively explored aerosol ERF uncertainty in a model that accounts for rapid atmospheric adjustments. Studies that do include rapid adjustments (e.g. Gettleman, 2015) rely on one-at-a-time experiments (where individual parameters or model structures are perturbed in isolation) which do a poor job of sampling the model uncertainty because they neglect important parameter interactions (Pianosi et al., 2016).

Here we present a perturbed parameter ensemble of the HadGEM3-GA4-UKCA global aerosol-chemistry-climate model and use model emulation (Lee et al., 2013) to enable the combined effects of uncertainties in 27 aerosol, cloud and other atmospheric model processes to be quantified. Compared to our previous studies (Carslaw et al., 2013; Regayre et al., 2014, 2015) we take a more holistic approach to exploring model forcing uncertainty here by accounting for both the uncertainty in cloud and other physical atmospheric processes, as well as the uncertainties in the aerosol component of the model. We also explore for the first time the uncertainty in aerosol ERF (including rapid atmospheric adjustments to aerosols), and in the components of ERF from aerosol-radiation interactions ($ERF_{ARI}$) and aerosol-cloud interactions ($ERF_{ACI}$). Other attempts to quantify the uncertainty in the ToA radiative flux caused by aerosols (Tett et al., 2013; Shiogama et al., 2012) explored only the current state of the atmosphere and not how it changes over time.



The main questions we address in this paper are: 1) How much of the uncertainty in aerosol ERF is caused by aerosol processes and how much by physical atmosphere processes? The answer is important because it will tell us how the tuning of model processes apparently unrelated to aerosols might inadvertently affect the aerosol ERF that models calculate. 2) What are the processes that cause uncertainty in the aerosol ERF and to what extent do they also affect the observable radiative state of the atmosphere? This is important because aerosol ERF uncertainty will only be effectively constrained by observations if the uncertainty in both the ERF and the observations are driven by the same uncertain processes (Lee et al., 2016). 3) How much does tuning the radiative state of the model (i.e., ruling out implausible model settings) affect the range of aerosol ERFs? The effect of tuning of, for example, ToA radiative flux (Lohmann and Ferrachat, 2010; Mauritsen et al., 2012) on the aerosol ERF is not normally considered. However, we show that many model variants (and parts of uncertain parameter space) can be ruled out using ToA flux observations and that such state variable observations can play an important part in reducing the overall uncertainty in aerosol ERF. The results from this paper inform our more comprehensive effort to constrain aerosol ERF uncertainty using multiple observational quantities (Johnson et al., 2018).

In Section 2 we outline our methodology, then in Section 3.1 we quantify the magnitude of the uncertainty in aerosol ERF, $ERF_{ARI}$ and $ERF_{ACI}$ through comprehensive sampling of model parameter uncertainty. We then analyse the main causes of uncertainty in aerosol ERF over multi-century and multi-decadal periods in Section 3.2 and the causes of ToA radiative flux uncertainty in Section 3.3 using sensitivity analysis techniques (Section 2). We also quantify the relative importance of atmospheric and aerosol parameters as sources of uncertainty in aerosol ERF and ToA radiative flux in Section 3.3. In Section 3.4 we identify the main causes of uncertainty in aerosol ERF and its components within 11 climatically important regions. Following Lohmann and Ferrachat (2010), we then explore how constraint of the model state using present-day ToA flux observations influences the plausible range of aerosol ERF (Sections 3.5.1 and 3.5.4). We show that while the relationships between the important driving parameters and individual parameter ranges are well constrained by ToA flux measurements (Sections 3.5.2 and 3.5.3), the range of credible aerosol ERFs is only moderately (10%) constrained. We investigate the causes of the modest constraint in sections 3.5.2, 3.5.3 and 4.

## 2    Methods

### 2.1    Set-up of the HadGEM-UKCA aerosol-climate model

We used the UK Hadley Centre Met Office Unified Model (HadGEM3, 2017) including release version 8.4 of the UK Chemistry and Aerosol (UKCA) model, within which the evolution of particle size distribution and size-resolved chemical composition of aerosols are calculated using the GLObal Model of Aerosol Processes (GLOMAP; Spracklen et al., 2005; Mann et al., 2010). The model has a $1.25° \times 1.875°$ horizontal resolution and 85 vertical hybrid pressure levels. The aerosol size distribution is defined by seven log-normal modes: one soluble nucleation mode as well as soluble and insoluble Aitken, accumulation and coarse modes. The aerosol chemical components are sulphate, sea salt, black carbon, particulate organic carbon and dust.



Secondary organic aerosol material is produced from the first stage oxidation products of biogenic monoterpenes under the as-
sumption of zero vapour pressure. After kinetic condensation onto existing aerosols organic aerosols (primary and secondary)
are treated as one chemical tracer.
The GLOMAP model resolves new particle formation, particle coagulation, gas-to-particle transfer, cloud processing (aque-
ous chemistry) and the deposition of gases and aerosols. Sulphate particles form by binary homogeneous nucleation (Vehkamäki
et al., 2002) throughout the atmosphere and by organically-mediated nucleation (Metzger et al., 2010) in the boundary layer.
The activation of aerosol particles into cloud droplets is calculated using distributions of sub-grid vertical velocities (West
et al., 2014) and the removal of cloud droplets by autoconversion into rain drops is calculated by the physical atmosphere
model. Aerosol removal by impaction scavenging of falling raindrops (within and below clouds) in the physical atmosphere
model depends partly on the collocation of clouds and precipitation (Boutle et al., 2014). Soluble particles grow according to
the relative atmospheric humidity using composition dependent hygroscopicity factors ($\kappa$) in accordance with 'Köhler theory'
(Petters and Kreidenweis, 2007).
Successive versions of the GLOMAP model have been widely evaluated against global measurements of particle number
concentration (Spracklen et al., 2010; Reddington et al., 2011), chemical compositions (Spracklen et al., 2011b; Schmidt et al.,
2011; Browse et al., 2012) and cloud active aerosol concentrations (Korhonen et al., 2008; Spracklen et al., 2011a; Pringle
et al., 2012). The HadGEM models are subject to constant monitoring for ongoing use in numerical weather prediction and
have informed successive Coupled Model Inter-comparison Project (CMIP) experiments (Taylor et al., 2012). HadGEM ca-
pably represents changes in cloud regime (Nam et al., 2012); one of the requirements for simulating rapid adjustments to
aerosol perturbations (Stevens and Feingold, 2009; Zhang et al., 2015). Cloud water responses to aerosols may be too strong
in the HadGEM model because the current model version does not represent enhanced drying in polluted clouds (Torrence and
Compo, 1998). However, over multiple cloud regimes the cloud water response is not of a sufficient magnitude to be climati-
cally important (Malavelle et al., 2017).
Anthropogenic emission scenarios prepared for the Atmospheric Chemistry and Climate Model Inter-comparison Project
(ACCMIP; Lamarque et al., 2010) and prescribed in some of the CMIP Phase 5 experiments (Taylor et al., 2012) are pre-
scribed here. Carbonaceous aerosol emissions from fires were prescribed using a ten year average of 2002 to 2011 monthly
mean data from the Global Fire and Emissions Database (GFED3; van der Werf et al., 2010).
Model horizontal winds were relaxed (nudged) towards winds from the European Centre for Medium-Range Weather Fore-
casts (ECMWF) ERA-Interim reanalysis above around 2 km. Nudging of atmospheric states is used primarily to evaluate
output from global models (Telford et al., 2008) or to ensure that pairs of simulations have near-identical atmospheric states,
so that aerosol and/or chemistry perturbations can be applied and their effects quantified using single realisations of each simu-
lation. In 'free-running' (non-nudged) simulations radiative fluxes need to be averaged over many decades in order to produce



signals stronger than the noise resulting from internal variability (Kooperman et al., 2012). Nudging to horizontal winds above
around 2 km forces synoptic-scale dynamical features to be consistent across the ensemble, whilst allowing boundary layer
atmospheric adjustments in response to changes in aerosols to be affected by the parameter perturbations.

Each simulation was subject to a seven-month spin-up period from a consistent starting simulation, with parameters set to
their median values for the first four months. Parameter perturbations were applied during the final three months of the spin-up
period, after which a full year of data was produced for each ensemble member. Aerosol ERF is calculated as the difference in
net ToA short-wave plus long-wave radiative fluxes between pairs of simulations with identical parameter settings but distinct
prescriptions of anthropogenic emissions (1850, 1978 and 2008). The aerosol ERF and its components were calculated based
on the method of Ghan (2013).

**2.2 Parameter sampling**
The 27 parameters perturbed in the ensemble, as well as the roles they play in the model, are presented in Table A1. We per-
turbed 9 parameters in the physical atmosphere model known to affect the properties and distribution of clouds and humidity
within the boundary layer (*atmospheric parameters*; Sexton et al. 2018) in combination with 18 aerosol emission, deposition
and process parameters (*aerosol parameters*) known to affect cloud droplet number concentrations (Lee et al., 2013) and/or
aerosol cloud-albedo effect forcing (the $\text{ERF}_{ACI}$ without accounting for rapid adjustments) at the global (Carslaw et al., 2013;
Regayre et al., 2014) and/or regional scale (Regayre et al., 2015). Some parameters have been included in the ensemble because
they represent model structural advances with inherent process uncertainty (Yoshioka et al., In prep.).

We did not attempt to include an exhaustive set of uncertain parameters in the experimental design. Current supercomputing
resources are too valuable to justify an uninformed, exhaustive exploration of model uncertainty. Instead, we used one-at-a-time
perturbation screening experiments (not shown) to identify the parameters most likely to influence radiative forcing within the
model. The parameters included in the preliminary screening process were identified by model experts as the key parameters
within individual model schemes (e.g. cloud microphysics) and/or model processes (e.g. cloud droplet activation) with the
potential to significantly affect aerosol ERF.

The parameters we perturb here are likely to have readily identifiable counterparts in other climate models. All global cli-
mate models have similarities because they describe the same physical processes and although process parametrisations can
differ between models they often share common biases when compared to measurements (Knutti et al., 2013). Therefore, our
aim to identify the main causes of aerosol ERF uncertainty in the HadGEM model (Section 3) will provide valuable clues for
reducing the aerosol ERF uncertainty in other models.



### 2.2.1 Definition of atmospheric parameters

**Rad_Mcica_Sigma**: *The fractional standard deviation of the sub-grid cloud condensate as seen by radiation*. This parameter controls the inhomogeneity of cloud condensate within vertically overlapping sub-grid clouds (Räisänen et al., 2004) which is used to calculate cloud radiative fluxes. Higher values of Rad_Mcica_Sigma increase cloud condensate inhomogeneity and hence reduce cloud albedo (because of the non-linear relationship between albedo and cloud condensate; Barker and Räisänen, 2005). Atmospheric temperature profiles respond to changes in the cloud radiative fluxes and can induce changes in precipitation rates and cloud amount. The effect of perturbing Rad_Mcica_Sigma on reflected radiation is largest in regions of persistent stratocumulus cloud where low-altitude, high-albedo clouds occupy a substantial fraction of each model grid box.

**C_R_Correl**: *Cloud and rain sub-grid horizontal correlation*. The collocation of clouds and rain within the model is important because it determines the accretion rate of cloud droplets and aerosols by rain drops. Higher values cause more accretion because regions of high cloud water are closely correlated with regions of high precipitation. Perturbations to this parameter affect cloud radiative properties by altering in-cloud interstitial aerosol concentrations and cloud amount.

**Niter_Bs**: *Number of microphysics iteration substeps*. The microphysical processing of in-cloud interstitial aerosols and cloud droplets is controlled by the cloud microphysics scheme within the physical atmosphere model. The values of this parameter determine the degree of processing within a model timestep. Each iteration of the microphysics scheme allows drops to grow larger before precipitation occurs. Therefore, higher parameter values allow for greater microphysical processing and cause the model to produce less light rain. This affects the amount of liquid water within clouds and alters the amount of cloud which is important for cloud radiative effects.

**Ent_Fac_Dp**: *Entrainment amplitude scale factor*. This convection scheme parameter controls the shape of the convective mass flux and the sensitivity of convection to relative humidity. Higher values reduce the depth of convection and suppress convective precipitation. This parameter is important for cloud radiative effects for several reasons. First, the retention of cloud water increases cloud amount and short-wave reflectivity. Second, lower altitude clouds have a higher cloud top temperature and attenuate less of the long-wave energy emitted by the Earth's surface. Third, if atmospheric moisture is not precipitated convectively, the increase in relative humidity causes more large-scale, frontal precipitation which affects spatial distributions of aerosols and clouds and hence the aerosol ERF.

**Amdet_Fac**: *Mixing detrainment rate scale factor*. This parameter controls the rate of humidification of the atmosphere and the shape of the convective heating profile. Amdet_Fac is important for cloud radiative effects for similar reasons to Ent_Fac_Dp. Both parameters affect clouds through their influence on convection but through different mechanisms. Higher values of Amdet_Fac increase atmospheric humidity and temperature leading to enhanced convection.



**Dbsdtbs_Turb_0**: *The cloud erosion rate*. This parameter alters the radiative properties of clouds by altering the rate at which unresolved sub-grid motions mix clear and cloudy air. Higher values cause more rapid mixing of clear, dry air into clouds, thereby reducing cloud liquid water content, autoconversion of cloud droplets to rain drops and cloud amount. The atmospheric lifetimes of aerosols and precursor gases are noticeably affected by this parameter.

**Mparwtr**: *Maximum value of the function controlling convective parcel maximum condensate*. Convective parcels near the Earth's surface precipitate when the amount of moisture reaches the threshold set by this parameter. Higher values increase cloud amount and lifetime by reducing convective precipitation. As with other convective parameters Mparwtr affects cloud radiative effects and aerosols by altering the spatial distributions of clouds and precipitation.

**Dec_Thres_Cld**: *The threshold for cloudy boundary layer decoupling*. Boundary layer stability plays an important role in determining the magnitude of cloud radiative effects because a well-mixed, stable boundary layer retains more heat and permits more dynamic activity. This parameter is the threshold at which the boundary layer decouples from the rest of the atmosphere. Hence, higher parameter values lead to a more well-mixed boundary layer, increased cloudiness and longer in-cloud processing times for aerosols.

**Fac_Qsat**: *Rate of change of convective parcel maximum condensate with altitude*. The maximum amount of moisture a convective parcel can hold transitions from the threshold set by the parameter Mparwtr at the surface to a much smaller threshold at high altitudes. Fac_Qsat controls the rate at which this threshold changes with altitude. Fac_Qsat therefore influences cloud radiative effects through similar mechanisms to Mparwtr (higher values suppress precipitation and increase cloud amount and lifetime) but is more important in the upper boundary layer.

### 2.2.2 Definition of the aerosol parameters

**Ageing**: *Ageing of hydrophobic aerosols*. Carbonaceous aerosols are assumed to be non-hygroscopic when emitted into the atmosphere and cannot act as cloud condensation nuclei until sufficient layers of sulphuric acid and condensible organic matter coat their surface. This parameter is the number of monolayers of soluble material required to convert initially insoluble aerosols into cloud condensation nuclei. Higher values reduce the conversion rate of hydrophobic to hygroscopic aerosols. This parameter is important for aerosol ERF because it affects cloud condensation nuclei and the removal rate of highly-absorbing carbonaceous aerosols from the atmosphere.

**Cloud_pH**: *pH of cloud droplets*. The pH of cloud droplets is used in the aqueous chemistry module of GLOMAP to calculate the conversion of $SO_2$ into sulphate particles. Cloud droplet pH depends on kinetic and thermodynamic processes that are not explicitly simulated. Therefore, we use a globally defined value of cloud droplet pH to control the reaction rate. Uncertainty in this parameter accounts for the simplification in its application. Higher values of this parameter increase sulphate





production near $SO_2$ emission sites and tend to reduce aerosol concentrations in remote regions (through effects on new particle formation). Therefore, the cloud pH parameter affects the spatial distribution of aerosols which is important for aerosol ERF.

**Carb_BB_Ems**: *Carbonaceous biomass burning emission scale factor*. Higher values of this scale factor increase the amount of carbonaceous aerosols emitted into the atmosphere from large-scale biomass burning. Carbonaceous aerosols are important for aerosol ERF because they absorb solar radiation and the resulting energy redistribution affects boundary layer temperatures and stability and can affect cloud cover (Gnanadesikan et al., 2017).

**Carb_BB_Diam**: *Carbonaceous biomass burning emission diameter (nm)*. This parameter determines the size of carbonaceous aerosols at time of emission. Higher values cause fewer, larger carbonaceous aerosols to be emitted for a given value of Carb_BB_Ems. Therefore, the total carbonaceous aerosol particle number is reduced, leading to fewer cloud condensation nuclei and a change in aerosol optical properties.

**Sea_Spray**: *Sea spray aerosol emission scale factor*. Aerosol ERF is sensitive to emission fluxes of natural aerosols because they strongly influence the pre-industrial background aerosol concentration and the relative magnitude of the change in aerosols over the industrial period. Perturbations to the wind-driven emission fluxes affect aerosol distributions in marine and coastal regions.

**Anth_SO2**: *Anthropogenic $SO_2$ emission scale factor*. $SO_2$ gas forms $H_2SO_4$ molecules which condense to form sulphate particles. Furthermore, $SO_2$ condenses onto existing particles increasing their size and solubility. Therefore, scaling anthropogenic $SO_2$ emissions affects aerosol ERF by influencing the concentrations and composition of present-day aerosols.

**Volc_SO$_2$**: *Volcanic $SO_2$ emission scale factor*. Volcanic $SO_2$ emissions are treated identically to anthropogenic $SO_2$ emissions. However, they are present in both the pre-industrial and present-day atmospheres so exert an influence on aerosol ERF through a similar mechanism as Sea_Spray by altering the pre-industrial aerosol concentration.

**BVOC_SOA**: *Biogenic secondary aerosol formation from volatile organic compounds scale factor*. Secondary organic aerosols form through multi-stage oxidation reactions of biogenic volatile organic compounds (monoterpenes in this case). This parameter scales the secondary organic aerosol emission flux, with higher values producing larger emissions. Perturbing this parameter changes the aerosol concentration and size distribution in the pre-industrial and present-day atmosphere.

**DMS**: *Dimethylsulphide surface ocean concentration scale factor*. Perturbing the concentration of DMS in the oceans alters the wind-driven flux of DMS into the atmosphere. DMS is important for aerosol ERF because it is a source of natural aerosols which affect the pre-industrial aerosol background concentrations. Similar to the Sea_Spray parameter, DMS affects aerosol concentrations in marine and coastal regions. However, marine DMS concentrations increase with ocean temperature so per-



turbations to this parameter will have the greatest influence on aerosol ERF in warmer months.

**Dry_Dep_Acc**: *Accumulation mode dry deposition velocity scale factor*. Aerosols are removed from the atmosphere at a velocity calculated using Brownian diffusion, impaction and interception. This calculation in the GLOMAP model depends on wind speeds and surface roughness. High values of this parameter more readily remove accumulation mode aerosols from the atmosphere causing a reduction in cloud condensation nuclei concentrations.

**Dry_Dep_SO2**: *$SO_2$ dry deposition velocity scale factor*. This parameter determines the removal of $SO_2$ gas from air masses that interact with the surface. The removal of $SO_2$ is important for aerosol ERF because $SO_2$ is a precursor for sulphate particles and condenses onto existing particles causing them to grow to the larger sizes needed to act as cloud condensation nuclei. Higher values of this parameter increase the removal rate of $SO_2$ from the atmosphere. This affects aerosol size distributions by simultaneously reducing particle formation rates and the growth rates of existing aerosols.

**Kappa_OC**: *Köhler coefficient of organic carbon*. Aerosol water uptake efficiency is determined by 'Köhler theory' using size and composition dependent hygroscopicity factors ($\kappa$; Petters and Kreidenweis, 2007). Higher values of this parameter increase the water uptake efficiency of the organic material in the particles. Perturbations to this parameter will change the light-scattering efficiency of the particles and the droplet activation process, thereby affecting cloud microphysical processes. In particular, cloud-active aerosol concentrations in the pre-industrial atmosphere are expected to be susceptible to this parameter value (Liu and Wang, 2010).

**Sig_W**: *Updraft vertical velocity standard deviation*. This parameter controls the width of the probability distribution of sub-grid vertical velocities used to calculate the activation of aerosols into cloud droplets. Higher Sig_W values widen the distribution of updraft velocities. The largest sub-grid updrafts within the distribution have the greatest influence on cloud droplet concentrations because, for any given supersaturation, a larger updraft velocity will cause a greater proportion of relatively small aerosols to activate. Higher values of Sig_W therefore increase cloud droplet concentrations, decrease precipitation efficiency (through reduced autoconversion rates), cloud liquid water content and cloud albedo. Sig_W perturbations have the greatest influence on cloud droplet concentrations in regions of relatively high aerosol concentrations because in such environments droplet activation is updraft-limited rather than aerosol-limited.

**Dust**: *Dust emission scale factor*. Dust aerosols are large, insoluble particles when emitted, but are treated as hygroscopic once sufficiently aged by the condensation of soluble material onto the particle surface (as defined by the 'ageing' parameter). We perturb dust emissions in our ensemble because they are important for the $ERF_{ARI}$ component of aerosol ERF. Furthermore, dust influences cloud-active aerosol concentrations (Manktelow et al., 2010) and cloud droplet concentrations (Karydis et al., 2017).





**Rain_Frac**: *Fraction of cloud-covered area in large-scale clouds where scavenging occurs*. Rain and clouds do not correlate perfectly (as discussed in the C_R_Correl definition). Higher values of this parameter allow aerosols to be scavenged by rain drops over a greater fraction of cloudy areas. The value of this parameter is important for aerosol ERF because it affects aerosol atmospheric lifetimes.

**Cloud_Ice_Thresh**: *Threshold of cloud ice fraction above which nucleation scavenging of aerosol material is suppressed*. The scavenging of aerosol material in dynamic rain systems is controlled partly by the rain formation process - either collision-coalescence process that efficiently removes many aerosol particles in raindrops or the Wegener-Bergeron-Findeisen process in mixed-phase clouds, which leads to less aerosol scavenging and seems to account for the efficient winter-time transport of aerosols to the Arctic (Barrett et al., 2011; Browse et al., 2012). In our previous studies (Regayre et al., 2014, 2015) we defined a temperature below which scavenging was suppressed. Here, we instead use the mass fraction of ice to define a threshold above which no nucleation scavenging occurs. Higher values require a greater proportion of ice to be present before scavenging is suppressed. This parameter is important for high latitude aerosol concentrations and cloud radiative effects (Browse et al., 2012; Regayre et al., 2015; Yoshioka et al., In prep.).

**BC_RI**: *Imaginary part of the black carbon refractive index*. This parameter controls the absorption of radiation as it passes through aerosols containing black carbon. Higher values of the imaginary refractive index cause more energy to be absorbed and re-emitted by black carbon aerosols. The real part of the refractive index is defined according to the imaginary part meaning that this parameter also controls the scattering of radiation by black carbon aerosols. Higher values of the real part cause more incoming radiation to be refracted towards the Earth's surface (more forward scattering). Perturbations to BC_RI affect $ERF_{ARI}$ as well as the vertical profile of atmospheric heating and hence convection, cloud amount and cloud radiative effects. Our simulations do not account for effect of depositing light-absorbing carbonaceous aerosols on snow (Bond et al., 2013), nor the air-sea interactions that enhance rapid adjustments in marine regions (Gnanadesikan et al., 2017).

**OC_RI**: *Imaginary part of the organic carbon refractive index*. The absorption of radiation by organic carbon is controlled by this parameter. Unlike BC_RI, the real part of the organic carbon refractive index is held constant. Therefore, perturbations to this parameter have no effect on the refractive properties of organic carbon. Otherwise, OC_RI affects the atmosphere through the same mechanisms as BC_RI.

One potentially important parameter that we did not perturb is the autoconversion rate of cloud droplets into rain drops (although we did perturb Rain_Frac and C_R_Correl which affect aerosol and cloud droplet removal by rain drops). The coupling between the GLOMAP model and the cloud microphysics scheme is currently one-way: cloud droplet concentrations calculated in GLOMAP are used in the autoconversion scheme and thereby affect precipitation rates, cloud liquid water content and albedo. However, precipitation only alters the cloud droplet concentrations in HadGEM and not aerosol concentrations within the GLOMAP model. For aerosol concentrations to be directly altered by the autoconversion process, the coupling would need



to be two-way so that cloud droplet concentrations in GLOMAP were consistent with those calculated in the atmospheric
model's microphysics scheme.

Other HadGEM simulations showed that over multiple cloud regimes cloud liquid water path is not substantially affected
by aerosols through autoconversion (Malavelle et al., 2017), suggesting that neglecting the uncertainty in this process is not
important to our results. However, in relatively polluted regions (such as the North Atlantic) cloud liquid water path responses
to aerosols in low-altitude clouds (particularly Stratocumulus) are likely to be overestimated in the model because of known
structural errors (Torrence and Compo, 1998). Despite this, the cloud liquid water path response to aerosols in low, warm
clouds is weaker in HadGEM than in other global climate models (Ghan et al., 2016). Therefore, autoconversion may be more
important in other models, but will likely be overstated (Torrence and Compo, 1998). This process should be considered in fu-
ture uncertainty analysis studies once shared model structural errors are addressed and the process uncertainty better quantified.

## 2.3 Statistical methodology

Maximin Latin Hypercube sampling was used to create a parameter combination design of 162 points with excellent space-
filling properties that provide information on model output across the 27-dimensional parameter uncertainty space. A sim-
ulation with all parameters set to their median values (from distributions described in table A1) was also included in the
ensemble. Emulators were then constructed which describe individual model outputs (e.g., ToA flux, aerosol ERF) over the
27-dimensional space of the uncertain parameters. Emulators provide a statistical representation of model output for all points
within the multi-dimensional parameter space and have been widely used to analyse climate models (Lee et al., 2013; Carslaw
et al., 2013; Tett et al., 2013; Regayre et al., 2014; Hamilton et al., 2014; Regayre et al., 2015; Johnson et al., 2015; Lee et al.,
2016) as well as complex models in many other areas of science, including hydrology (Liu and Gupta, 2007), galaxy formation
(Rodrigues et al., 2017) and disease transmission (Andrianakis et al., 2017).

In total 217 perturbed parameter simulations were created for each anthropogenic emission period including a set of 54
simulations with parameter combinations that augment the original design and were used to validate the emulators. Twenty-six
simulations did not complete an annual cycle in at least one of the anthropogenic emission periods (1850, 1978 and 2008)
because the combinations of parameters caused the model to fail. Hence, the ensemble of simulations for each period was
made up of the remaining 191 simulations. Once emulators were validated, new emulators conditioned on output from the 191
perturbed parameter simulations (with better space-filling properties) were created by combining the validation simulations
with the original set of simulations.

Probability density functions (pdfs; Table A1) were used to represent expert beliefs about parameter uncertainty. We pre-
dominantly used trapezoidal distributions (Hetzel, 2012) to represent parameter uncertainty in order to avoid having an overly-





centralized multi-variate sample (Yoshioka et al., In prep.).

By combining perturbed parameter ensembles with model emulation and then densely sampling emulator output using the
extended-FAST sampling method (Saltelli et al., 1999), we were able to perform sensitivity analyses (Saltelli et al., 1999, 2000;
Lee et al., 2012) and decompose the variance in model output into individual components. We used the percentage reduction
in variance which would be achieved if a parameter value was known exactly as our main statistic for identifying the causes of
uncertainty. Emulation and sensitivity analyses were applied at the individual model gridbox level (degraded to N96 model res-
olution) as well as at the regional and global mean level. For the sensitivity analyses, samples of 270000 members were drawn
from the emulators at parameter combinations determined by the parameter pdfs. The sensitivity analysis results are therefore
informed by expert knowledge about the model behaviour in relation to the uncertain processes. However, for the constraint of
aerosol ERF using ToA flux observations we sampled one million model variants using uniform pdfs. This sampling approach
uses the expert-elicited parameter pdfs to determine the ranges of uniform pdfs for sampling but neglects expert prior beliefs
about parameter value likelihoods. As such, the effects of applying the observational constraint and expert knowledge can be
quantified and compared. Furthermore, the effect of applying the observational constraint on the uncertain parameter space can
be more readily assessed when uniform pdfs are used to create the original sample because parameter combinations are more
evenly spaced throughout the 27-dimensional parameter space.

Preliminary parameter combination screening tests revealed that values of Ent_Fac_Dp higher than around 1.8 in combi-
nation with values of Amdet_Fac higher than around 8.0 caused model simulations to fail. This part of the 27-dimensional
parameter space (a corner of a 2D plane) was removed from the ensemble design and analyses. The sampling method used to
perform the sensitivity analyses, was adapted to reject samples from the 2D corner of parameter space not included in the de-
sign. Rejected combinations of the Ent_Fac_Dp and Amdet_Fac parameters were re-sampled from the restricted 2D parameter
space without affecting the sampling frequency across the remaining 25-dimensional parameter space.

## 424   3   Results

### 425   3.1   Uncertainty in aerosol ERF and its components

Figure 1 shows pdfs of the global mean aerosol ERF (from 1850 to 2008) and its components; $ERF_{ARI}$ and $ERF_{ACI}$. The 95%
credible interval of aerosol ERF used in the sensitivity analysis is -2.18 to -0.71 $\mathrm{W\,m^{-2}}$. Most of the uncertainty in aerosol
ERF comes from the $ERF_{ACI}$ component, which has a credible interval of -2.20 to -0.61 $\mathrm{W\,m^{-2}}$ and captures much of the
recognised uncertainty in this forcing term (Myhre et al., 2013; Shindell et al., 2013). We also account for above-cloud aerosols
(Ghan, 2013) in our calculation of $ERF_{ACI}$ and $ERF_{ARI}$ which affects the balance between these two components of aerosol
ERF (Yoshioka et al., In prep.). This adjustment results in distributions of weaker $ERF_{ARI}$ values and stronger $ERF_{ACI}$ values



in our sample compared to (Myhre et al., 2013). We discuss these effects further in Section 3.1.2.

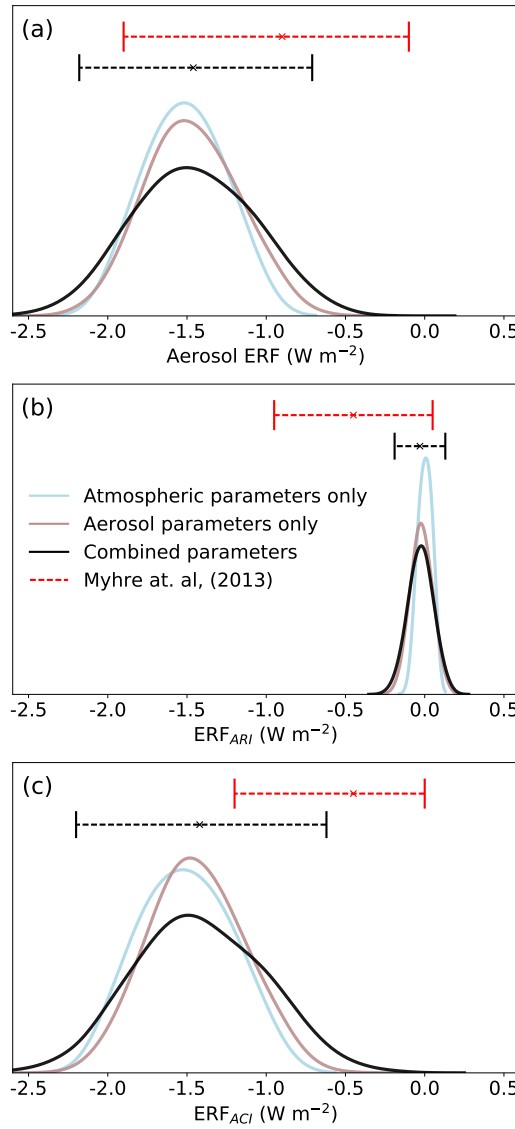

**Figure 1.** Probability density functions of 1850-2008 (a) aerosol ERF, (b) $ERF_{ARI}$ and (c) $ERF_{ACI}$. Each sample contains 270000 emulator-derived model variants informed by the expert-elicited prior probability distributions of parameter values. Samples with aerosol and atmospheric parameter uncertainties neglected (Table A1) were obtained by setting each neglected parameter to its median value in the corresponding pdf. 90% credible intervals from (Myhre et al., 2013) are presented as red horizontal lines with best estimates marked using crosses. Our 95% credible intervals are presented in black and the sample median is presented using a cross.



The sample of aerosol ERFs in Fig. 1 has already been constrained by our choice of probability distributions for the uncer-
tain parameters (Table A1). When we use uniform parameter distributions to sample parameter combinations (Section 2.3) the
credible range (95%) of aerosol ERFs is -2.65 to -0.68 $\mathrm{W\,m^{-2}}$. By applying expert beliefs about parameter value likelihoods
the aerosol ERF credible range is only -2.18 to -0.71 $\mathrm{W\,m^{-2}}$ (Fig. 1(a)). This implies that by applying the combined knowledge
of experts with an understanding of the model processes and parametrisations we have effectively reduced the aerosol ERF
credible range by around 25%.

The strongest aerosol ERFs in our distribution would lead to a negative forcing when combined with best estimates of
changes in other forcing agents over the industrial period. A net negative forcing is incompatible with the observed increase in
global mean surface temperatures over the industrial period (e.g. HadCRUT4, 2017). However, there is substantial uncertainty
in the ERFs of multiple other forcing agents (Myhre et al., 2013; Fig. 8.16 and 8.18) so our most negative aerosol ERF values
cannot be considered implausible using this criteria. Structural aspects of the model could account for the strongest forcings.
For example, our model is missing marine sources of organic aerosols and related processes (Gantt et al., 2015) which, if in-
cluded, would act as an important source of ice-nucleating particles (Vergara-Temprado et al., 2017) and pre-industrial aerosols
(Gordon et al., 2017) which would weaken the aerosol forcing (Carslaw et al., 2013). However, our perturbed parameter ranges
were to some extent intended to encompass the uncertainty caused by those structural deficiencies we were aware of. The
values in the tails of the aerosol ERF pdf are likely to be the result of setting multiple parameters important for aerosol ERF
to extreme values, which are also likely to cause extreme present-day ToA flux values and be considered implausible when
compared to observations (Sections 3.5).

Figure 1 also shows the separate effects of the 18 combined aerosol parameters and the 9 combined physical atmosphere
model uncertainties. Neglecting the uncertainty in aerosol parameters (by setting them to their median values in the all model
variants) results in a 95% credible aerosol ERF interval of -1.98 to -1.04 $\mathrm{W\,m^{-2}}$, while neglecting uncertainty in atmospheric
parameters results in a credible interval of -2.00 to -0.90 $\mathrm{W\,m^{-2}}$. Summary statistics of forcing from these samples are pre-
sented in Table A2. The distribution of aerosol ERF (as well as $\mathrm{ERF}_{ARI}$ and $\mathrm{ERF}_{ACI}$) is wider and flatter (has a larger
variance) in the combined sample than the distributions of atmosphere-only and aerosol-only sampled values. This suggests
that important interactions between atmospheric and aerosol parameters cause the most extreme aerosol ERF values. The ef-
fects of the aerosol and physical model uncertainties do not have an additive effect on the aerosol ERF uncertainty because
of compensating effects between the groups of parameters. These results show that both atmospheric and aerosol parameter
perturbations are required to comprehensively sample model uncertainty. The main atmospheric and aerosol sources of aerosol
ERF uncertainty are identified in Section 3.2.1.





### 3.1.1  Uncertainty in ERF$_{ACI}$

Maps of the means and standard deviations of ERF$_{ACI}$ resulting from perturbations to our 27 atmospheric and aerosol parameters are presented in Fig. 2. Forcings stronger than -3.5 W m$^{-2}$ are concentrated over anthropogenic aerosol sources (particularly Asia, America and Europe) and in marine stratocumulus regions (Atlantic Ocean, North Pacific Ocean and the South Pacific Ocean off the South American coast). The standard deviation of ERF$_{ACI}$ is largest (up to 6 W m$^{-2}$) in the same regions and is typically of the same order of magnitude as the mean regional value. The spatial distribution of mean ERF$_{ACI}$ is very similar to the Atmospheric Chemistry and Climate Model Intercomparison Project (ACCMIP) multi-model mean pattern (Shindell et al., 2013). However, the magnitudes of forcing differ, particularly over remote marine regions. For example, our mean ERF$_{ACI}$ is stronger than -5 W m$^2$ over much of the North Pacific Ocean, whereas the ACCMIP mean aerosol ERF in the Pacific is stronger than -3.5 W m$^2$ only in coastal regions near to anthropogenic sources. These strong remote marine ERF$_{ACI}$ values go some way to explaining the differences in global mean ERF$_{ACI}$ between our sample (around -1.4 W m$^2$) and the ACCMIP multi-model mean (around -0.9 W m$^2$). In part, the magnitude of our ERF$_{ACI}$ values are caused by the above-cloud aerosol adjustment (Ghan, 2013). Our model has a relatively weak cloud liquid water path response to aerosols (Ghan et al., 2016; Malavelle et al., 2017), which suggests that our very negative marine forcing values are not caused by an overly strong aerosol second indirect effect.

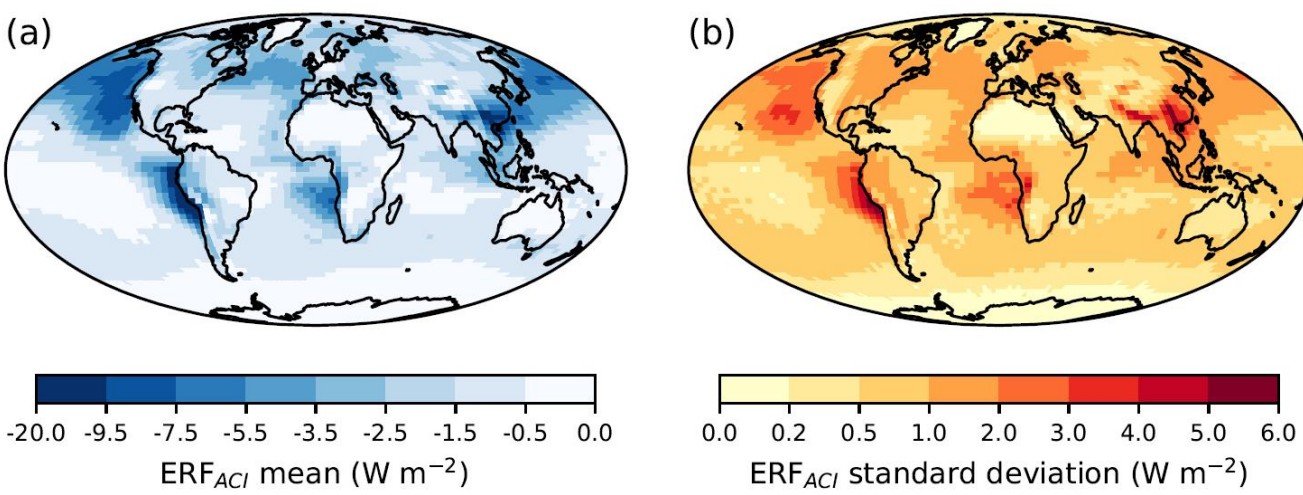

**Figure 2.** (a) Mean and (b) standard deviation for 1850-2008 ERF$_{ACI}$ forcing. Values were calculated using output from 270000 emulator-derived model variants at the individual pixel level once degraded to N96 model resolution. These samples of model variants are informed by the expert-elicited parameter pdfs.



### 3.1.2 Uncertainty in $\text{ERF}_{ARI}$

Fig. 3 shows the spatial pattern of mean $\text{ERF}_{ARI}$ and its standard deviation. Global mean $\text{ERF}_{ARI}$ is near zero (95% credible range -0.19 to 0.13 $\text{W m}^{-2}$; Fig. 1; Table A2). Although the possibility of a globally positive $\text{ERF}_{ARI}$ has previously been considered unlikely (Boucher et al., 2013), it has important implications for our understanding of interactions between absorbing aerosols, cloud-processes and boundary-layer dynamics. The near-zero global mean $\text{ERF}_{ARI}$ results from the cancellation of positive and negative regional forcings. Positive mean $\text{ERF}_{ARI}$ values (up to 10 $\text{W m}^{-2}$) occur in regions where carbonaceous aerosols often overlie relatively high-albedo clouds (continental Asia and off the west coasts of Africa and South America). It is in these regions that the standard deviation of $\text{ERF}_{ARI}$ is also largest (up to 5 $\text{W m}^{-2}$). Light-absorbing aerosols above cloud heat the local atmosphere, which can suppress convection and affect cloud cover. This is important for calculating the $\text{ERF}_{ARI}$ from our simulations because we account for above-cloud scattering and absorption of aerosols in line with Ghan (2013). Neglecting the effects of above-cloud aerosols in the $\text{ERF}_{ARI}$ produces no positive values for this forcing component (95% credible interval -0.69 to -0.24; Yoshioka et al., In prep.). Therefore, the magnitude of $\text{ERF}_{ARI}$ over Asia, Africa and South America (where it is positive and reduces cloud cover) determines the sign of global mean $\text{ERF}_{ARI}$.

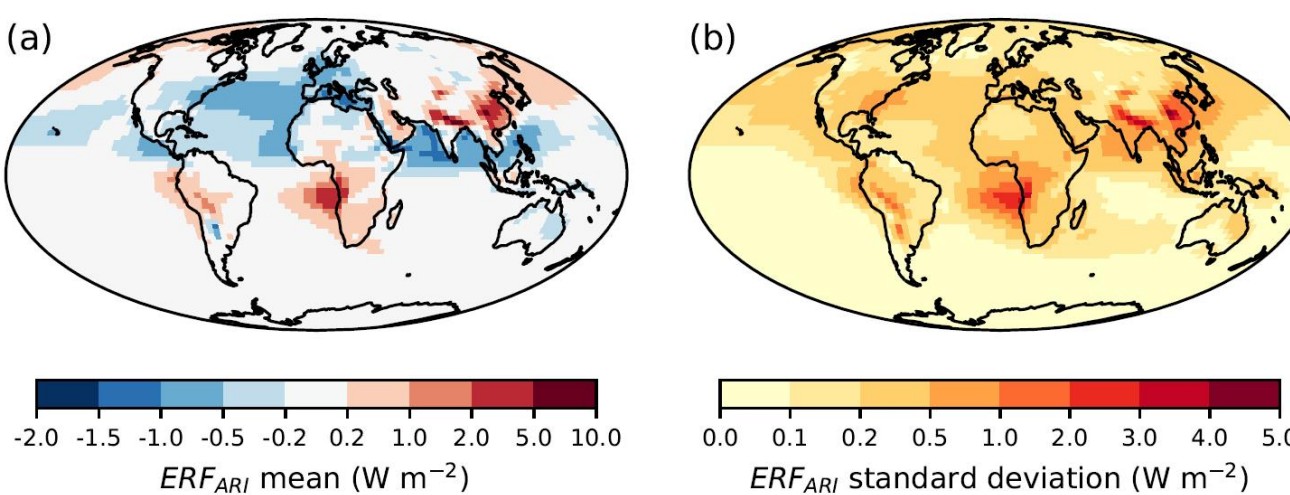

**Figure 3.** (a) Mean and (b) standard deviation of 1850-2008 $\text{ERF}_{ARI}$ forcing. Values were calculated using output from 270000 emulator-derived model variants at the individual pixel level. These samples of model variants are informed by the expert-elicited parameter pdfs.



### 3.2 Sources of uncertainty in aerosol ERF and its components

#### 3.2.1 Sources of uncertainty in global mean ERF$_{ACI}$

Fig. 4 summarises the causes of variance (sometimes referred to as the 'main effects') in global mean ERF$_{ACI}$, ERF$_{ARI}$ and aerosol ERF. Natural aerosol emissions (here, Sea_Spray, DMS and BB_Diam) persist as important sources of industrial-period ERF$_{ACI}$ uncertainty, as in previous studies of several climate models (Wilcox et al., 2015) and the aerosol-only component of a global model (Carslaw et al., 2013). However, by far the largest source of uncertainty is the Rad_Mcica_Sigma parameter. This cloud radiation parameter affects the spatial homogeneity of simulated clouds, altering (amongst other things) reflected radiation, tropospheric temperature profiles and cloud amount. Therefore, by altering the radiative *state* of clouds in the pre-industrial and present-day atmospheres Rad_Mcica_Sigma affects uncertainty in the simulated *change* in cloud radiative state (the ERF$_{ACI}$). Model process parameters Sig_W and C_R_Correl cause uncertainty in ERF$_{ACI}$ by altering the efficiency of the cloud droplet activation and deposition processes respectively. Other parameters cause a small amount of the ERF$_{ACI}$ uncertainty but only in individual months. Therefore, the six parameters and associated processes identified here are the key to understanding the uncertainty in the global, annual mean ERF$_{ACI}$ in HadGEM.

#### 3.2.2 Sources of uncertainty in global mean ERF$_{ARI}$

The sources of global mean ERF$_{ARI}$ variance are summarised in Fig. 4(b). Parameters related to the emission and radiative properties of carbonaceous absorbing aerosols (BC_RI, OC_RI and BB_Ems) are amongst the largest sources of ERF$_{ARI}$ variance in all months. However, the emission flux of carbonaceous aerosols (BB_Ems) and the radiative properties of organic carbonaceous aerosols (OC_RI) cause much more of the ERF$_{ARI}$ variance in high emission months (Jun - Aug) than they do in the annual mean. In other months with lower concentrations of carbonaceous aerosols, uncertainty in anthropogenic emissions (here, Anth_SO2) is the largest source of global mean ERF$_{ARI}$ variance. Anthropogenic emissions affect the ERF$_{ARI}$ by influencing aerosol properties in the present-day atmosphere. Other parameters (notably, Rad_Mcica_Sigma and Sig_W) affect the balance between ERF$_{ACI}$ and ERF$_{ARI}$ by altering cloud radiative properties which are important for calculating above-cloud aerosol effects (Ghan, 2013). Rad_Mcica_Sigma and Sig_W are the only parameters identified as important causes of uncertainty in both ERF$_{ACI}$ and ERF$_{ARI}$.

#### 3.2.3 Sources of uncertainty in industrial-period global mean aerosol ERF

The aerosol ERF is the sum of the ERF$_{ACI}$ and ERF$_{ARI}$. Therefore, the sources of aerosol ERF variance are also sources of variance in the forcing components. The causes of aerosol ERF variance are summarised in Fig. 4(c). Aerosol ERF shares more sources of variance with ERF$_{ACI}$ than with ERF$_{ARI}$ because ERF$_{ACI}$ is the stronger and more uncertain forcing component (Fig. 1). Natural aerosol emissions (Sea_Spray, DMS and BB_Diam) and model process parameters (Sig_W and C_R_Correl)







**Figure 4.** Percentage contributions to variance in global, monthly and annual mean 1850-2008 (a) $ERF_{ACI}$, (b) $ERF_{ARI}$ and (c) aerosol ERF. Each bar contains only those parameters that cause at least 3% of the variance and interactions between parameters are neglected. Therefore, the percentage of variance accounted for is less than 100%. The monthly and annual median values and 95% credible intervals (from the 270000 model variants) are displayed in the top panel. The monthly median values are connected in bold and the credible intervals are shaded gray.

collectively cause over half of the aerosol ERF variance. Each of these key parameters causes a similar proportion of the
aerosol ERF and $ERF_{ACI}$ variances. However, the cloud radiation parameter (Rad_Mcica_Sigma) causes more of the $ERF_{ACI}$



variance (around 35%) than aerosol ERF variance (less than 30%), despite also causing around 25% of the $\text{ERF}_{ARI}$ variance.
This suggests that the $\text{ERF}_{ACI}$ and $\text{ERF}_{ARI}$ responses to Rad_Mcica_Sigma are of opposite sign and thus partially cancel in
the aerosol ERF calculation. The other main difference between sources of aerosol ERF and $\text{ERF}_{ACI}$ variance comes from
anthropogenic emissions. Anthropogenic emission uncertainty (Anth_SO2) causes up to 10% of the aerosol ERF variance in
all months. However, Anth_SO2 only causes a small percentage of the $\text{ERF}_{ACI}$ variance in a few months. Therefore, this
parameter's contribution to aerosol ERF variance is predominantly through its influence on the $\text{ERF}_{ARI}$ component of forcing.

### 3.2.4 Sources of uncertainty in multi-decadal aerosol ERF

The causes of aerosol radiative forcing uncertainty are known to depend on the anthropogenic emission period examined
(Carslaw et al., 2013; Regayre et al., 2014). A more detailed understanding of the causes of uncertainty in aerosol ERF re-
quires sensitivity analyses over multiple time periods. In this section, we examine the pattern of uncertainty in multi-decadal
(1978-2008) aerosol ERF, identify the main causes of uncertainty in multi-decadal aerosol ERF and discuss how these results
inform our understanding of aerosol ERF on longer time scales.

Fig. 5 shows the spatial pattern of mean aerosol ERF and its standard deviation over the 1978-2008 period. Global anthro-
pogenic sulphate emissions peaked in the late 1970s (Lamarque et al., 2010) then decreased in Europe and North America as
a result of clean air legislation, but increased significantly in Asia (Smith et al., 2011). Therefore, there are distinct regions
of positive and negative aerosol ERF in the 1978-2008 period. The cancellation of the regional aerosol ERFs of opposite sign
cause a near-zero global mean aerosol ERF (95% credible range of -0.6 to 0.8 $\text{W m}^{-2}$). Over continental land masses, the
aerosol ERF standard deviation is largest (between 0.5 and 5 $\text{W m}^{-2}$) in regions of substantial mean aerosol ERF (absolute
mean larger than around 1 $\text{W m}^{-2}$). The aerosol ERF standard deviation is larger than around 0.3 $\text{W m}^{-2}$ over most marine
regions and is largest over regions of persistent stratocumulus cloud, even when the mean forcing is near-zero (e.g. off the west
coast of South America). This suggests the sign of recent-decadal aerosol ERF forcing is uncertain over those regions.

The sources of variance in aerosol ERF and its components over the 1978 to 2008 period are summarised in Fig. 6. The
sign of the aerosol ERF over the 1978-2008 period is uncertain for much of the year and is only definitively negative in the
Northern Hemisphere summer. The cancellation of positive and negative regional aerosol ERFs has three main implications
for the global mean sensitivity analysis. Firstly, not all of the causes of regional aerosol ERF will be evident in the global mean
analysis (Regayre et al., 2015). Nevertheless, the causes of uncertainty in global mean 1978-2008 aerosol ERF will inform
our understanding. Secondly, the causes of global mean aerosol ERF uncertainty are seasonally dependent because changes in
the magnitude of incoming solar radiation determine the relative importance of regional uncertainties. Thirdly, the competing
regional effects cause the total variance accounted for by individual parameters to be much less than 100% (as low as 55% in
some months) with many parameters causing only a small amount (around 5%) of the variance. This suggests that important
interactions between multiple parameters in multiple regions are causing much of the global mean aerosol ERF variance in





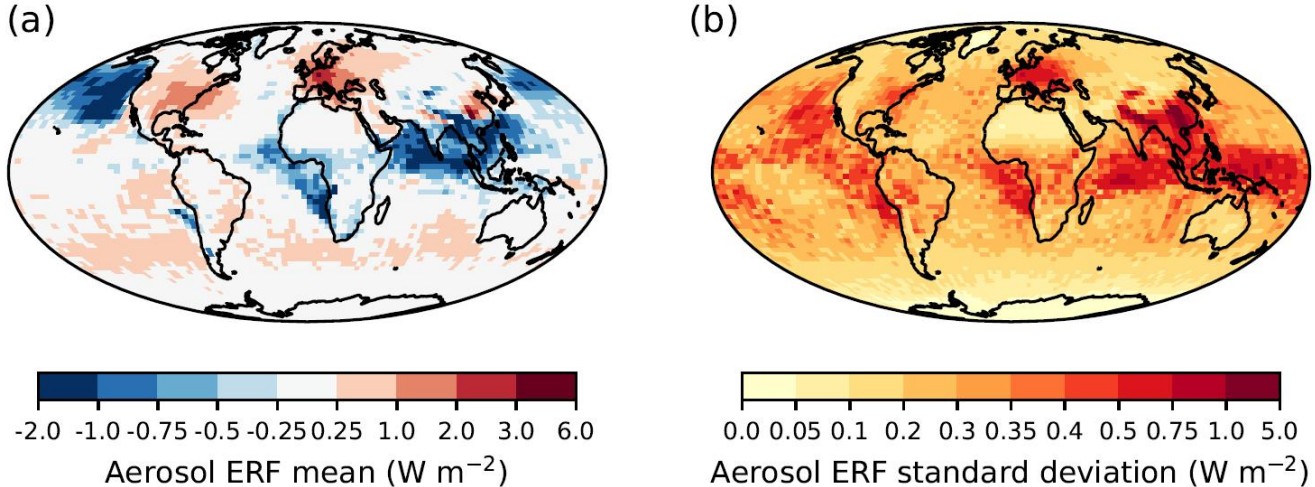

**Figure 5.** (a) Mean and (b) standard deviation of 1978-2008 aerosol ERF. Values were calculated using output from 270000 emulator-derived model variants at the individual pixel level. These samples of model variants are informed by the expert-elicited parameter pdfs.

recent decades.

565        There are multiple ways in which the causes of aerosol ERF uncertainty in the 1978-2008 period differ from those in the

1850-2008 period. Firstly, natural aerosol emission parameters have little influence on recent-decadal aerosol ERF uncertainty
because the global mean 1978-2008 aerosol ERF depends more linearly on changing anthropogenic emissions than the 1850-
2008 aerosol ERF (Carslaw et al., 2013). Secondly, the cloud radiation parameter Rad_Mcica_Sigma causes very little (less
than 3%) of the 1978-2008 aerosol ERF variance. The reduced importance of this parameter as a cause in aerosol ERF un-
certainty results from the cancellation of regional aerosol ERFs of opposite sign, which also depends on the linearity of the
multi-decadal aerosol ERF response to anthropogenic emission changes. Thirdly, in the 1978-2008 period anthropogenic and
model process parameters are a larger source of aerosol forcing uncertainty, as in previous analysis of this period (Regayre
et al., 2014). Here, uncertainty in the deposition rates of aerosols and aerosol precursor gases account for most (around 20%
each) of the multi-decadal aerosol ERF variance. The aerosol process parameter Cloud_pH causes another 10% of the 1978-
2008 aerosol ERF variance. The anthropogenic emission parameter Anth_SO2 and other model process parameters (Sig_W,
Rain_Frac, and BC_RI) each cause only a small amount (around 3%) of the variance.

**3.3    Sources of uncertainty in ToA radiative flux**
Identifying the sources of ToA reflected shortwave radiation (RSR) uncertainty will inform our understanding of how radiative
flux measurements can help to constrain the aerosol ERF uncertainty (Lohmann and Ferrachat, 2010) because the aerosol ERF





**Figure 6.** Percentage contributions to variance in 1978-2008 global, monthly and annual mean (a) ERF$_{ACI}$, (b) ERF$_{ARI}$ and (c) aerosol ERF. Figure features are identical to Fig. 4.

is essentially the aerosol-forced change in RSR between the pre-industrial (or 1978) and present-day atmospheres (plus addi-
tional small changes in outgoing long-wave radiation). The causes of present-day ToA RSR variance are summarised in Fig. 7
and are very similar in the pre-industrial and 1978 atmospheres (not shown).

The dominant source of ToA RSR uncertainty is the cloud radiation parameter Rad_Mcica_Sigma, which was also the
dominant parameter for the pre-industrial to present-day aerosol ERF. Uncertainty in this parameter alone causes over 60% of





**Figure 7.** Percentage contributions to variance in present-day (2008) global, monthly and annual mean ToA (a) cloudy-sky RSR, (b) clear-sky RSR and (c) RSR. Figure features are identical to Fig. 4.

the RSR variance by altering the total cloud albedo. The dominant role of this cloud radiative parameter in causing uncertainty
in the ToA radiative flux and aerosol ERF suggests that constraining this parameter to a very narrow range should constrain the
uncertainty in radiative fluxes (Haerter et al., 2009; Lohmann and Ferrachat, 2010) and consequentially in aerosol ERF (Lee
et al., 2016). But of course, there are a number of other parameters (Dbsdtbs_Turb_0, Ent_Fac_DP, Sig_W and C_R_Correl)
that cause ToA RSR uncertainty by altering the amount and/or albedo of clouds in the model. The mechanisms for altering



cloudiness and therefore the ToA radiative flux are different for each parameter. The Dbsdtbs_Turb_0 parameter causes around
10% of the ToA RSR variance by altering the mixing rate of clean and cloudy air masses. Increasing the proportion of dry
air in clouds has a dramatic effect on the amount of low-altitude cloud simulated in the model, making Dbsdtbs_Turb_0 the
dominant cause of uncertainty in low-altitude cloud-fraction (Fig. 8). The Ent_Fac_Dp, Sig_W and C_R_Correl parameters
each cause around 5% of the RSR variance. The Ent_Fac_Dp parameter affects the strength of convection which also alters
precipitation rates and the vertical distribution of simulated clouds. Sig_W controls the activation of cloud condensation nuclei
into cloud droplets (affecting droplet effective radius and cloud albedo) and C_R_Correl alters the rate of cloud droplet accre-
tion by precipitating rain drops. The only parameter to cause ToA RSR variance (around 10%) by directly altering atmospheric
aerosol concentrations is Sea_Spray.

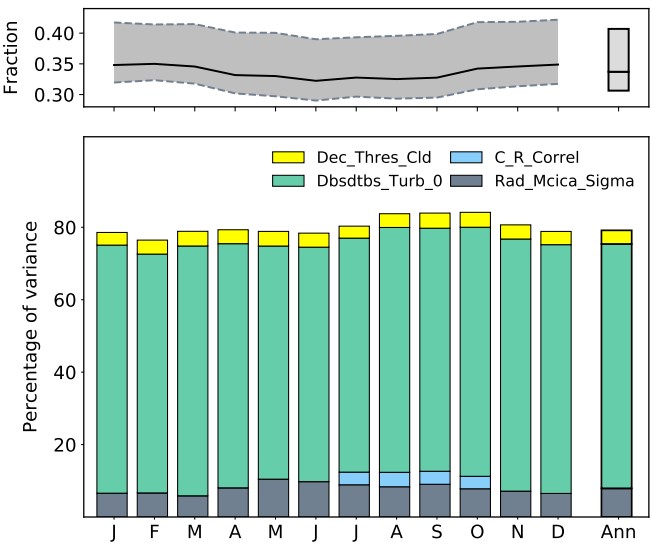

**Figure 8.** Percentage contributions to variance in present-day (2008) global, monthly and annual mean low altitude cloud amount. Figure features are identical to Fig. 4.

Figure 9 summarises the relative contributions of atmospheric and aerosol parameters to uncertainty in global mean values
of present-day ToA RSR (from Fig. 7) and aerosol ERFs over the periods 1978-2008 (Fig. 6) and 1850-2008 (Fig. 4). Atmo-
spheric parameters cause the majority (around 80%) of the variance in present-day ToA radiative flux, but only around 30%
of the variance in 1850-PD aerosol ERF, and less than 10% of the 1978-PD aerosol ERF variance. The rest of the uncertainty
is attributable to the aerosol model. This disparity arises because contributions to variance in aerosol ERF depend on how
parameters influence the atmosphere's response to the *change* in anthropogenic emissions, while RSR variance depends on
how they influence the *state* of the atmosphere.





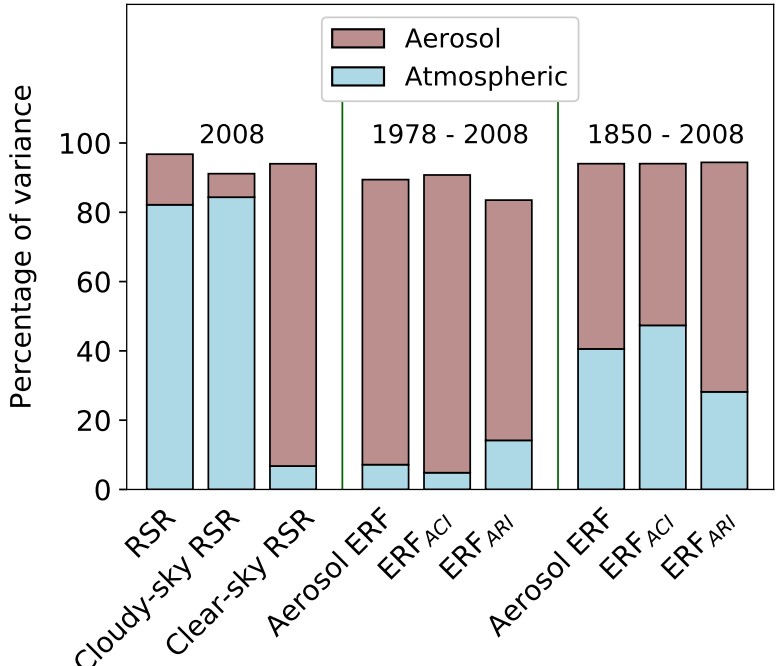

**Figure 9.** The relative contributions from atmospheric and aerosol parameters to variance in ToA radiative fluxes and aerosol effective radiative forcing over the 1978-2008 and 1850-2008 periods.

## 3.4 Identifying the sources of uncertainty at the regional level

### 3.4.1 Regional sources of uncertainty

Regional forcings can be important drivers of global and regional climate change (Chalmers et al., 2012; Booth et al., 2012; Bollasina et al., 2013; Shindell et al., 2013; Kirtman et al., 2013; Villarini and Vecchi, 2013; Allen et al., 2014). Furthermore, important sources of aerosol forcing uncertainty may be overlooked if regional sensitivity analysis results are neglected (Regayre et al., 2015). Examining how these sources of regional forcing uncertainty combine to cause uncertainty in global mean forcing uncertainty will inform our understanding of how to best observationally constrain the uncertainty. We identified regions of substantial aerosol ERF (stronger than around -2.5 $\mathrm{W\,m^{-2}}$) for more in-depth analysis (Table A3 and Fig. 10). The corresponding sources of aerosol ERF variance in each region are summarised in Fig. 11.





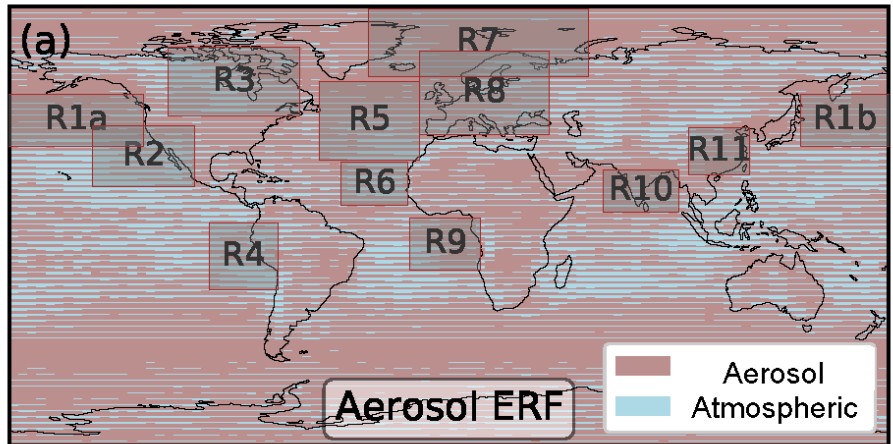

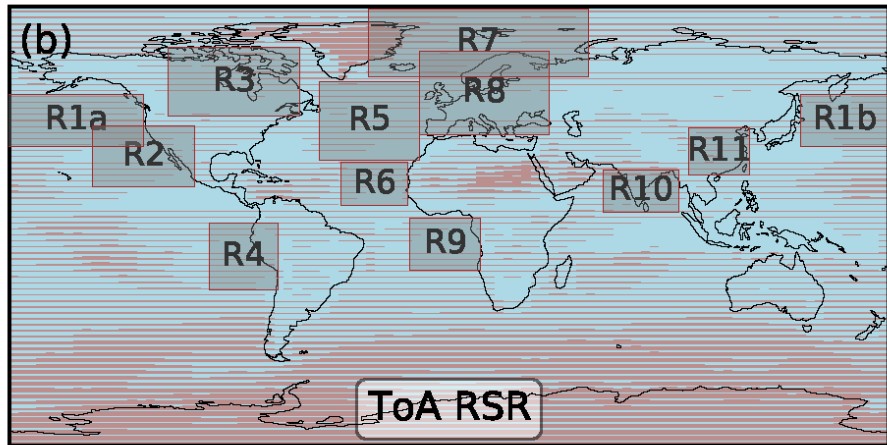

**Figure 10.** Maps of contributions to variance in (a) 1850-2008 aerosol ERF and (b) present-day (2008) ToA RSR from atmospheric and aerosol parameters. Each pixel contains a box that is shaded in proportion to the amount of variance caused by each source of uncertainty.





**Figure 11.** Sources of variance ((a), (b) and (c)) and grouped atmospheric and aerosol contributions to variance ((d), (e) and (f)) for 1850-2008 annual mean (a),(d) aerosol ERF, (b),(e) ERF$_{ARI}$ and (c),(f) ERF$_{ACI}$ for the 11 regions defined in table A3 and highlighted in Fig. 10.

The main causes of regional aerosol ERF uncertainty are often those parameters that cause global mean uncertainty. How-
ever, there are substantial differences between regions. Some parameters are important causes of global mean aerosol ERF
uncertainty because they cause a small amount (at least 5%) of the uncertainty in nearly all regions. For example, the DMS
parameter causes around 5% of the aerosol ERF variance in most regions and consequentially causes around 15% of the global
mean variance. The cloud radiation parameter Rad_Mcica_Sigma (which causes nearly 30% of the industrial-period aerosol
ERF variance) also causes aerosol ERF variance in most regions. But, the amount of regional aerosol ERF variance accounted



for by this parameter ranges from less than 3% (R9) to around 35% (R2, R3, R4).

Other parameters are important causes of global mean aerosol ERF uncertainty despite being important causes of uncertainty
in only around half of the regions examined. For example, the Sea_Spray parameter (which causes nearly 20% of the global
mean aerosol ERF variance) is by far the largest source (around 60%) of aerosol ERF variance in the North Pacific (R1) and
causes between 10 and 30% of the variance in several other marine regions. However, in tropical marine regions (R6 and R10)
and regions containing continental land mass (R3, R8 and R11) Sea_Spray causes less than 3% of the aerosol ERF variance.
The land-based regions (R3, R8, R11) are also where the cloud updraft parameter Sig_W causes aerosol ERF variance. The
importance of Sig_W over continents suggests cloud albedo is most sensitive to uncertainty in updraft velocity in the most-
polluted regions where cloud droplet concentrations are updraft-limited (Reutter et al., 2009; Sullivan et al., 2016).

Anth_SO2 makes its greatest contribution to aerosol ERF uncertainty in tropical marine regions (R6 and R10) by causing
uncertainty in $\text{ERF}_{ARI}$. The Anth_SO2 parameter also causes up to 40% of the $\text{ERF}_{ARI}$ variance near anthropogenic sources
(R3 and R8) and up to 30% in outflow regions (R1, R2, R5). However, these substantial causes of $\text{ERF}_{ARI}$ variance translate
into small (less than 10%) causes of aerosol ERF variance in most regions. The aerosol deposition parameter (Dry_Dep_Acc)
also causes more of the regional $\text{ERF}_{ARI}$ variance (up to 45%) than regional aerosol ERF variance (less than 15%). However,
despite being an important cause of 1850-2008 aerosol ERF uncertainty in several regions, the dry deposition parameter is not
an important cause of global mean aerosol ERF uncertainty over this period (Fig. 4(c)).

The importance of carbonaceous aerosol parameters (Carb_BB_Ems, Carb_BB_Diam, BC_RI and OC_RI) as causes of
aerosol ERF uncertainty are highly region dependent. Uncertainty in the emission flux of carbonaceous aerosols Carb_BB_Ems
causes between 25 and 45% of the $\text{ERF}_{ARI}$ variance in and near biomass burning regions (R4, R7 and R9). However, this only
translates into a cause of aerosol ERF uncertainty in regions R7 and R9 where the Carb_BB_Ems parameter also causes uncer-
tainty in $\text{ERF}_{ACI}$ . Uncertainty in the size of emitted carbonaceous absorbing aerosols (Carb_BB_Diam) is more important as
a cause of uncertainty in $\text{ERF}_{ACI}$ than in $\text{ERF}_{ARI}$ because it determines the capacity for carbonaceous aerosols to act as cloud
condensation nuclei. Therefore, Carb_BB_Diam predominantly causes aerosol ERF variance (up to 15%) in the cloudiest
regions (R1, R5, R6 and R9). Uncertainty in the radiative properties of carbonaceous aerosols (BC_RI and OC_RI) collec-
tively cause $\text{ERF}_{ARI}$ variance in almost all regions. However, uncertainty in aerosol ERF is affected by these parameters only
over China (R11) and near to India (R10). Over China the anthropogenic emission parameter (Anth_SO2) is surpassed by the
BC_RI and OC_RI parameters as causes of $\text{ERF}_{ARI}$ and aerosol ERF uncertainty, despite carbonaceous aerosols making up
a relatively small proportion of aerosol emissions in these regions (Granier et al., 2011). The BC_RI parameter causes around
50% of the $\text{ERF}_{ARI}$ variance and around 25% of the variance in aerosol ERF in China. However, anthropogenic emissions
do cause uncertainty in $\text{ERF}_{ARI}$ and aerosol ERF in the Pacific (an outflow region for Chinese emissions). Near India, uncer-
tainty in BC_RI and OC_RI cause around 30% and 10% of the aerosol ERF variance respectively and cause a smaller amount
(between 5 and 10%) of variance in each of the forcing components. Despite being important sources of forcing uncertainty at



the regional level, Carb_BB_Diam is the only parameter related to carbonaceous aerosols which causes uncertainty in global,
annual mean aerosol ERF.

Figure 11 (d)-(f) shows that atmospheric parameters combined can cause up to around 50% of the regional aerosol ERF vari-
ance despite causing only around 30% of the global mean aerosol ERF variance. However, there are multiple regions where
uncertainty in the physical atmosphere parameters causes less than 20% of the aerosol ERF variance. Where atmospheric pa-
rameters are an important source of regional aerosol ERF uncertainty, the Rad_Mcica_Sigma parameter is almost always the
most important. On its own uncertainty in Rad_Mcica_Sigma causes over 20% of the aerosol ERF variance in coastal Pacific
regions (R2 and R4) as well as continental regions (R3 and R8). The atmospheric parameter controlling the accretion rate
of aerosols by rain drops (C_R_Correl) causes around 10% of the aerosol ERF variance in several tropical or sub-tropical
regions off the western coast of continents (R2, R4, R6 and R9). These are all regions of persistent stratocumulus cloud where
cloud albedo is highly susceptible to changes in aerosol concentrations and size distributions. The clear- and cloudy- air mix-
ing parameter Dbsdtbs_Turb_0 causes between 5 to 10% of the variance in aerosol ERF and its components in the Northern
Hemisphere regions of persistent stratocumulus cloud (R2 and R6) but not in Southern Hemisphere regions (R4 and R9). This
suggests that the relatively polluted Northern Hemisphere stratocumulus clouds are more sensitive to the sub-grid mixing of
clear- and cloudy air masses. In tropical regions (R6, R9 and R10) the convective parameter Mparwtr causes a small amount
(3 to 5%) of the aerosol ERF variance. This parameter alters the timing of precipitation and therefore affects cloud and aerosol
amount, and the $ERF_{ACI}$ near the equator where convective instability and precipitation are greatest. The regions where phys-
ical atmosphere parameters cause the least aerosol ERF variance are either near to anthropogenic emission sources (R9, R10
and R11) or downwind of them (R1 and R7).

These results show that the relative importance of individual parameters as sources of uncertainty differ between regions.
However, the most important causes of global mean aerosol ERF uncertainty also cause uncertainty at the regional level.

**3.5    Observational constraint of the aerosol ERF uncertainty**
**3.5.1    Effect of ToA RSR constraint on aerosol ERF uncertainty**
We now explore the extent to which present-day measurements of global mean ToA RSR could in principle help to constrain
the change in flux between two time periods (the aerosol ERF), which was previously explored by Lohmann and Ferrachat
(2010) who perturbed four physical atmosphere parameters. We expect some constraint of aerosol ERF uncertainty based on
the common causes of uncertainty in ToA RSR and aerosol ERF. Observational constraint of a model output variable can lead
to constraint of the uncertain parameters. Therefore, when two model output variables share common causes of uncertainty
we can expect that constraint of one output will lead to constraint of the other. Our approach of drawing large samples of one
million parameter combinations from model emulators (using uniform pdfs for each parameter; Section 2.3) enables this link





through the uncertain parameters to be understood, which is not possible just from a perturbed parameter ensemble alone (e.g.
Lohmann and Ferrachat, 2010).

Our analysis reveals substantial overlap in the combinations of parameters causing uncertainty in 1850-2008 aerosol ERF
and present-day ToA RSR. The parameters Rad_Mcica_Sigma, Sea_Spray, C_R_Correl and Sig_W account for about 60%
of the aerosol ERF uncertainty and about 80% of the ToA flux uncertainty. It is important to note that it is irrelevant for
the observational constraint process that the ToA flux is much larger than the aerosol ERF. The important factor is that their
uncertainties are caused by common uncertain parameters, so constraint of one of them will constrain the other through the
constraint of the plausible parameter ranges and relationships.

Figure 12 shows the effect of constraining the modelled present-day RSR to within $\pm 0.25 \mathrm{W\,m^{-2}}$ of 98.3 $\mathrm{W\,m^{-2}}$, the
multi-year average of observations from the Clouds and the Earth's Radiant Energy System (CERES; Loeb et al., 2009). The
$\pm 0.25 \mathrm{W\,m^{-2}}$ represents within-CERES product uncertainty (Loeb et al., 2012) and neglects multiple other sources of satellite
observational uncertainty (Loeb et al., 2009; Hartmann et al., 2013). We also neglect uncertainty caused by unknown model
structural errors (Goldstein and Rougier, 2004; Sexton et al., 2012; Stier et al., 2013), observation representativeness errors
(Schutgens et al., 2017) and the emulators themselves (Oakley and O'Hagan, 2004). Therefore, our RSR observational con-
straint provides an upper bound on the potential reduction in aerosol ERF uncertainty. This tight constraint eliminates 97% of
the model variants and the observationally constrained RSR range is less than 2% of the original unconstrained range. Conse-
quently, the smaller set of model variants also predicts reasonably constrained 1978 and 1850 RSR ranges. However, despite
reducing the plausible parameter space by 97% and the RSR range by 98% the impact on the aerosol ERF uncertainty is more
modest. The effect of applying the RSR observational constraint is to rule out 1850-2008 aerosol ERF values lower than around
-2.4 $\mathrm{W\,m^{-2}}$, which represents around 15% of the original aerosol ERF range (the 95% credible range is reduced by around
10%; Table A4). This reduction in aerosol ERF range is much less than the 56% reduction found by Lohmann and Ferrachat
(2010) based on a set of 169 perturbed parameter simulations (compared to our one million model variants). We discuss the
reasons for this modest reduction in aerosol ERF uncertainty in sections 3.5.2 and 3.5.3.





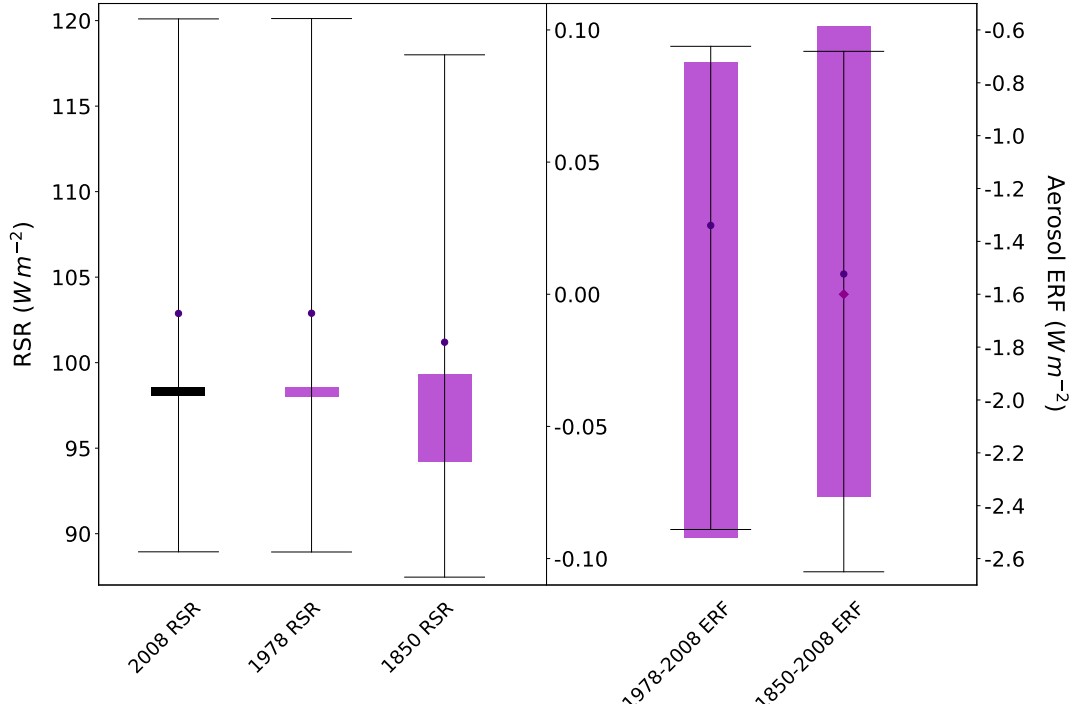

**Figure 12.** Observationally constrained present-day ToA RSR and the values of 1978 and 1850 ToA RSR and 1978-2008 and 1850-2008 aerosol ERF values from matching model variants. For each output variable the black lines show 95% credible intervals of the unconstrained one million member sample of model variants. The black box contains all model variants within $\pm 0.25\,\mathrm{W\,m^{-2}}$ of the CERES-observed global annual mean present-day ToA RSR value. Purple boxes represent the 95% credible intervals of values obtained using model variants (parameter combinations) in the observationally constrained sample. The middle and right-hand axes are for 1978-2008 and 1850-2008 aerosol ERF respectively. Output from the simulation with all parameters set to their median values are shown as dots. The median 1850-2008 aerosol ERF from the observationally constrained sample is displayed as a diamond.

### 3.5.2 Constraining the relationships between the aerosol ERF and uncertain parameters

Fig. 13 shows how aerosol ERF is related to the values of the four main causes of aerosol ERF uncertainty before and after applying the observational constraint. There are clear relationships between the aerosol ERF and the individual parameters, but they are highly uncertain (even in the constrained sample) because there are many compensating errors among the other parameters (i.e., many ways to combine the parameters to get the same ToA RSR but very different aerosol ERF; (Fig. 12). This diversity of credible model variants would be overlooked had we perturbed parameters individually, as is the case with one-at-a-time perturbation experiments (e.g. Gettleman, 2015).





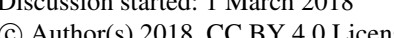

**Figure 13.** Probability density distributions of aerosol ERF and the parameters Rad_Mcica_Sigma, Sea_Spray, DMS, Sig_W in the unconstrained sample (first column; (a)-(d)) and in the sample constrained to match the observed global annual mean RSR (second column; (e)-(h)). Probability density distributions of parameter values are shown for the constrained sample ((i)-(n)). Colour bars labelled (a)-(n) correspond with the sub-figures and show the percentage of each sample within each pixel. Some colour bars apply to multiple panels.

For each of the one million model variants in our unconstrained sample individual parameter values were drawn from uni-
form distributions with ranges defined by the expert elicited pdfs. Therefore, prior to applying the observational constraint the
model variants were evenly dispersed across every two-dimensional parameter subspace. We are therefore able to quantify the
effect of the observational constraint on the plausibility of individual and combined parameter values.



The cloud radiation parameter Rad_Mcica_Sigma is negatively correlated with the ToA radiative flux and this leads to a pos-
itive correlation with aerosol ERF: Increasing its value decreases the simulated cloud albedo and hence the ToA RSR. But, ToA
RSR is more sensitive to Rad_Mcica_Sigma in the present-day atmosphere than in the pre-industrial (because higher aerosol
concentrations increase the cloud albedo) , so increasing the parameter value weakens the aerosol ERF. Figure 13 shows that
low values of Rad_Mcica_Sigma (less than around 0.4 in the scaled range 0 to 1) are inconsistent with the observed RSR. The
proportion of model variants with Rad_Mcica_Sigma values less than 0.4 drops from 40% in the unconstrained sample to just
8% in the constrained case. In other words, the observational constraint suggests the plausible lower limit of this parameter is
higher than we assumed in our expert elicitation. We can therefore state that the strongest aerosol ERFs are also implausible
(as shown in Fig. 12) because they are associated with low values of Rad_Mcica_Sigma.

Figure 13 also shows that the aerosol ERF is weaker for larger Sea_Spray and DMS values. An abundance of natural
aerosols increases background (pre-industrial) atmospheric aerosol concentrations and limits the influence of anthropogenic
aerosol emissions on clouds and radiation (Carslaw et al., 2013). Figure 13 shows that the observed ToA radiative flux is more
consistent with low emissions of natural aerosols (the density of Sea_Spray values larger than 0.5 decreases from 50% to
44% after constraint of the ToA flux). The decreased likelihood of higher natural aerosol emissions in the constrained sample
suggests that the weakest aerosol ERF values are less congruent with observed ToA RSR. However, there remain many obser-
vationally plausible model variants with high natural aerosol emissions.

The largest values of the cloud updraft parameter (Sig_W) are also less plausible in the constrained sample (Fig. 13; 27%
of the sample are larger than 0.7, instead of 30%). This suggests that present-day RSR observations are more consistent with
lower vertical velocities, but the largest values cannot be ruled out completely because of the way that other compensating
parameters affect RSR. Lower values of Sig_W weaken the aerosol ERF by reducing cloud droplet concentrations primarily in
the present-day polluted atmosphere, because cloud droplet activation is more sensitive to Sig_W in the present-day atmosphere
than the in the aerosol-limited pre-industrial atmosphere. Therefore, observational constraint of ToA radiative flux reduces the
likelihood of weak aerosol ERFs through a constraint of the distribution of Sig_W values.

**3.5.3    Constraining the relationships between uncertain parameters**
Fig. 13 also shows the important parameter inter-dependencies revealed by observationally constraining the ToA radiative flux.
The Rad_Mcica_Sigma and Sea_Spray parameters are positively correlated in the observationally constrained sample. For
example, a modelled ToA RSR consistent with observations can be achieved using high values of Rad_Mcica_Sigma (which
decreases cloud albedo) and relatively high values of Sea_Spray (which increases cloud albedo). In other words, these param-
eters have compensating effects on the ToA radiative flux. The same compensation applies to the aerosol ERF: the weakest
ERFs in our pdf (larger than around -1 $\mathrm{W\,m^{-2}}$) are associated with high Rad_Mcica_Sigma and high Sea_Spray values. How-
ever, the RSR and aerosol ERF depend on these two parameters in quite different ways. Higher Rad_Mcica_Sigma values



*weaken* the aerosol ERF by *reducing* the present-day ToA RSR, whilst higher Sea_Spray values *weaken* the aerosol ERF by
*increasing* present-day RSR. Hence, constraining the relationship between the two largest sources of aerosol ERF uncertainty
using observations of present-day RSR has not drastically reduced the aerosol ERF uncertainty.

The cloud droplet activation parameter (Sig_W) is also positively correlated with Rad_Mcica_Sigma in the observationally
constrained sample (Fig. 13). As with sea spray emissions, higher values of Sig_W increase cloud albedo and compensate
for the effect of high Rad_Mcica_Sigma values on ToA RSR. These parameters both exert a greater influence on present-
day cloud radiative properties; in the case of Sig_W, cloud radiative properties are more susceptible to this parameter in the
present-day simulations because cloud droplet activation is more likely to be updraft-limited (rather than aerosol-limited) in
an anthropogenically-polluted atmosphere. Therefore, in contrast to the Sea_Spray and Rad_Mcica_Sigma relationship, the
Sig_W and Rad_Mcica_Sigma parameters have additive (not compensating) effects on aerosol ERF. Parameters with additive
effects on the aerosol ERF are more susceptible to the effects of model equifinality. Therefore, the relationship between aerosol
ERF and Sea_Spray is better constrained than the relationship between aerosol ERF and Sig_W.

The Sig_W and Sea_Spray parameters both act to counter the effect of Rad_Mcica_Sigma on cloud albedo in the con-
strained sample. Therefore, the density of model variants with simultaneously large (above around 0.6) Sig_W and Sea_Spray
values is lower in the constrained sample (down from 16% to 11%). No such restrictions apply to simultaneously small values
(less than 0.4) of these two parameters. In fact the proportion of simultaneously small Sig_W and Sea_Spray in the sample
increases from 16% to 19% after applying the constraint which rules out other parts of parameter space. This suggests that in
simulations with low natural aerosol emissions and low cloud droplet activation efficiency, there are multiple other contributing
factors keeping the ToA RSR in agreement with observations. For example, by limiting the mixing rates of clear and cloudy air
masses, a low value of the Dbsdtbs_Turb_0 parameter (an important source of ToA RSR uncertainty) can compensate for the
decrease in cloud droplet concentrations caused by a low value of the cloud droplet activation parameter. A replacement source
of aerosols large enough to act as cloud condensation nuclei is also required to compensate for low natural aerosol emissions.
There are multiple ways in which this could be achieved. For example, a low value of the dry deposition velocity parameter
Dry_Dep_Acc (known to be important for cloud active aerosol concentrations; Lee et al., 2013) increases the atmospheric
lifetime of aerosols, allowing them to grow in size and activate to form cloud droplets, even in a low activation efficiency
simulations.

The DMS parameter has no obvious relationships with the other main sources of aerosol ERF uncertainty in the constrained
sample. This is despite DMS affecting aerosol ERF in the same regions as other key parameters and causing aerosol ERF
uncertainty through a similar mechanism to Sea_Spray. In other words, higher values of DMS and Sea_Spray suppress the
aerosol ERF by increasing background (1850) aerosol concentrations. Therefore, the value of the DMS parameter is more
likely (54% of the time) to be small (lower than 0.5) when the value of Sea_Spray is high (above 0.8). In summary, model
variants with high values of both of the important natural aerosol emission parameters are less likely to be consistent with the




observed ToA RSR.

These results highlight the importance of understanding the potential causes of equifinality when interpreting results from
such a complex model (Beven and Freer, 2001). Reducing the remaining uncertainty in global mean aerosol ERF will require
observations which further constrain the relationships between aerosol ERF and the key sources of uncertainty.

**3.5.4    Regional constraint of global mean aerosol ERF uncertainty**
Our overall aim is to constrain the uncertainty in global annual mean aerosol ERF because the total ERF is commonly used to
quantify the multi-model diversity in historically forced changes to the climate (Myhre et al., 2013; pp661). However, regional
variations in aerosol forcing can be important drivers of climate variability (Chalmers et al., 2012; Booth et al., 2012; Bollasina
et al., 2013; Shindell et al., 2013; Kirtman et al., 2013) and can contribute to global mean forcing uncertainty in complex
ways (Regayre et al., 2015). Therefore we now use satellite observations of the North Pacific (region R1; latitude 32N-54N;
longitude 125W-144E) ToA RSR from July to further constrain annual, global mean aerosol ERF uncertainty.

The regionally averaged CERES observed ToA RSR is 162.8 $\mathrm{W\,m^{-2}}$ (CERES, 2017) with an estimated uncertainty of $\pm2\%$
(Hartmann et al., 2013, 2.3.1, pp181). The original sample of one million model variants is reduced to around 10% by applying
the North Pacific July mean RSR constraint and to just 0.5% of the original sample by applying both the global mean and
North Pacific constraints together (Table A4). In combination with the global mean observation, the North Pacific RSR con-
straint has little additional effect on the credible forcing ranges (-2.30 to -0.56 $\mathrm{W\,m^{-2}}$ compared to -2.37 to -0.59 $\mathrm{W\,m^{-2}}$). The
range of plausible aerosol ERF values has been further reduced by only around 2%. This suggests that the regional observation
has provided little additional constraint on the relationships between aerosol ERF and the main sources of uncertainty (Fig. S1).

**4    Conclusions**
We sampled the uncertainty in 18 aerosol and 9 atmospheric parameters within a single global climate model, identified the
important causes of aerosol ERF uncertainty and constrained this uncertainty using ToA radiative flux measurements. The
credible range of aerosol ERF values in our original sample of one million model variants was -2.65 to -0.68 $\mathrm{W\,m^{-2}}$ when we
assume the parameter values have equal likelihood of being at any point in the elicited ranges. The aerosol ERF uncertainty de-
creases when we constrain global mean ToA RSR (-2.37 to -0.59 $\mathrm{W\,m^{-2}}$) and when we constrain both North Pacific and global
RSR (-2.30 to -0.56 $\mathrm{W\,m^{-2}}$). However, a greater reduction (25%) in the aerosol ERF uncertainty (95% credible range, -2.18 to
-0.71 $\mathrm{W\,m^{-2}}$) can be achieved by applying probability distributions to the parameters based on expert elicitation. These results
suggest that the strongest aerosol ERF values (about 20% of the unconstrained range) can be considered implausible based on



expert opinion and observational evidence.

Our results reveal that aerosol parameters take a dominant role over atmospheric parameters as the leading cause of aerosol ERF uncertainty over the industrial period and in recent decades. Atmospheric parameters cause the majority (over 80%) of the uncertainty in present-day ToA reflected short-wave radiation but only around 30% of the aerosol ERF variance. A handful of the aerosol and atmospheric parameters that we have examined dominate the uncertainty in global mean aerosol ERF. A cloud radiation parameter, natural aerosol emissions and model process parameters that affect cloud droplet formation and removal are the key sources of global mean aerosol ERF uncertainty over the industrial period. The most important causes of 1978-2008 aerosol ERF uncertainty are model process parameters controlling the deposition rates of aerosols and aerosol precursor gases. Our analysis shows that uncertainties in aerosol parameters are of secondary importance for determining present-day ToA radiative flux, but they are a much more important source (over half) of the uncertainty in the change in atmospheric radiative balance (the aerosol ERF) on multi-century and multi-decadal timescales.

Uncertainty in the $\text{ERF}_{ARI}$ component of forcing (-0.19 to 0.13 $\text{W}\,\text{m}^{-2}$) is largely caused by parameters related to carbonaceous aerosols. However, these parameters contribute little to uncertainty in the total aerosol ERF, which is dominated by uncertainty in the $\text{ERF}_{ACI}$ component of forcing (-2.20 to 0.61 $\text{W}\,\text{m}^{-2}$) in our analyses. In our simulations light-absorbing aerosols heat the local atmosphere above clouds, suppress convection and affect cloud cover. However, we do not represent all of the processes that determine the magnitude of carbonaceous aerosol forcing. For example, we neglect the deposition of absorbing-aerosols onto high-albedo land surfaces. Therefore, despite the large uncertainties in our carbonaceous aerosol parameters, our global mean $\text{ERF}_{ARI}$ uncertainty range does not span the range of values found by Bond et al., 2013.

At the regional level, uncertainty in aerosol ERF is predominantly caused by the same parameters that cause global mean aerosol ERF uncertainty. Some parameters such as the cloud radiation parameter Rad_Mcica_Sigma and the natural aerosol emission parameter DMS are important for global mean aerosol ERF uncertainty because they cause at least a small amount (5%) of the uncertainty in nearly all regions. Other important causes of global mean aerosol ERF uncertainty (Sea_Spray, Sig_W and Anth_SO2) are amongst the largest causes of the aerosol ERF uncertainty in some regions (marine, polluted and polluted-marine regions respectively) but cause very little of the uncertainty elsewhere. We show that because carbonaceous aerosols only cause aerosol ERF uncertainty in high-emission months and in regions close to emission sources, most of the carbonaceous aerosol parameters (with the exception of Carb_BB_Diam) are not important for global, annual mean aerosol ERF uncertainty.

A well-constrained multi-decadal historical aerosol ERF would provide more policy-relevant information on near-term temperature change than industrial-period ERF which remains challenging to constrain (Hawkins et al., 2017). Constraining recent-decadal aerosol ERF uncertainty may prove to be an easier task than constraining uncertainty in industrial-period forcing because the multi-decadal uncertainty is caused by model process parameters that could be observed directly. Global





mean aerosol ERF in recent decades depends more linearly on changing anthropogenic emissions than industrial-period aerosol
ERF. Therefore, the causes of aerosol ERF uncertainty in recent decades (1978-2008) are model deposition rates (model pro-
cess parameters) and anthropogenic emissions, whilst the 1850-2008 aerosol ERF is most sensitive to natural aerosol emissions.
The magnitude of global mean aerosol forcing on the decadal timescale depends on the combination of uncertain positive and
negative regional forcings (Regayre et al. 2015; Fig. 5). Hence, projects designed to improve our understanding of the state
and behaviour of aerosol-cloud-radiation interactions on regional scales and within specific cloud regimes will aid efforts to
constrain global mean forcing. In summary, reducing the uncertainty in aerosol ERF will require a much deeper understanding
of how the uncertainties in state variables, model parameters and the relationships between them combine at the regional and
global levels in complex global climate models. We develop our understanding of the potential to constrain regional aerosol
ERF uncertainty using multiple observable quantities in Johnson et al. (2018).

Climate models are routinely tuned to match present-day ToA radiative fluxes (in conjunction with multiple other obser-
vational metrics) so as to ensure accurate characterisation of the state of the atmosphere (Kay et al., 2012; Mauritsen et al.,
2012; Flato et al., 2013; Hourdin et al., 2017). Our sensitivity analysis shows that the ToA radiative flux and the 1850-2008
aerosol ERF share common sources of uncertainty. Therefore, observational constraint of ToA flux representing just 0.5% of
the model's prior range has reduced the 95th percent credible interval of our simulated global mean aerosol ERF by around
10%. These results counter the belief that observations of ToA reflected short-wave radiation should not constrain the aerosol
ERF (because RSR values are an order of magnitude larger than the aerosol ERF). However, comprehensively sampling model
uncertainty provides a densely populated multi-dimensional parameter space which connects the observed value (RSR) to the
model variable of interest (the aerosol ERF). The RSR observation constrains the parameter space and in doing so constrains
the aerosol ERF uncertainty. However, we caution that the constraint will only be robust if all relevant parameters affecting
RSR have been explored.

Our results show that the plausible ranges of individual parameters as well as the relationships between them are constrained
by present-day observations, thereby substantially reducing the model parameter space that can be considered observationally
plausible. We use RSR observations with a small observational uncertainty to demonstrate their potential use as a constraint
on aerosol ERF uncertainty. However, despite a very large reduction in plausible parameter space, the effectiveness of the
observational constraint is modest because it is hampered by compensating effects between multiple uncertain parameters. The
challenge now is to find optimum combinations of constraints that overcome this problem using a more robust framework that
accounts for all quantifiable sources of uncertainty (Sexton et al., 2012; Williamson et al., 2013). For aerosol ERF this means
simultaneously constraining aerosols, clouds, and radiation *state* variables as well as the relationships between them so as to
constrain uncertainty in the *change* of state on multiple timescales.

By highlighting how different parameters and processes control the change in planetary radiative balance in a single state of
the art model, our results suggest that compensating effects between groups of uncertain parameters and associated processes





are one important reason why uncertainty in aerosol ERF has persisted through several generations of climate model develop-
ment. Given the huge range of interacting processes and uncertainties, it is highly unlikely that single observational constraints
(as employed in so-called emergent constraint studies; e.g. Cherian et al. (2014)) will enable a robust reduction in aerosol
ERF uncertainty. Our results, combined with those of other studies that have comprehensively sampled model uncertainties
(Calisto et al., 2014; Lee et al., 2016; Ghan et al., 2016), suggest that reducing aerosol ERF uncertainty further will require the
simultaneous application of a large number of observational constraints (Sanderson, 2010; Sexton et al., 2012; Collins et al.,
2012; Reddington et al., 2017) covering polluted and pristine environments (Carslaw et al., 2013; Hamilton et al., 2014) and
targeting the specific processes and relationships identified here.

## 5 Code availability

Code can be made available upon request from the corresponding author.

## 6 Data availability

Data can be made available upon request from the corresponding author. The authors welcome use of the perturbed parameter
ensemble for advancing climate research.
*Author contributions.* L. Regayre tested the model configuration, designed and prepared the ensemble and analysed the results. L. Regayre
and K. Carslaw wrote the article. All authors contributed to the analysis and interpretation of results. K. Pringle, M. Yoshioka, L. Regayre,
K. Carslaw, J. Johnson and N. Bellouin helped prepare the model configuration that served as the template for the ensemble. L. Regayre
and J. Johnson designed the experiments. All simulations were created by L. Regayre. M. Yoshioka advised on computational aspects of the
ensemble creation. The screening of atmospheric parameters was conducted by L. Regayre, D. Sexton and K. Carslaw. L. Regayre and J.
Johnson elicited probability density functions of all aerosol parameters and K. Carslaw, N. Bellouin, K. Pringle, M. Yoshioka and L. Lee
participated (alongside many other experts) in the formal elicitation process.
*Competing interests.* The authors declare that they have no conflict of interest.
*Acknowledgements.* L. Regayre was funded by a Natural Environment Research Council (NERC) Doctoral Training Grant, and a CASE
studentship with the UK Met Office Hadley Centre. B. Booth was supported by the Joint UK DECC/Defra Met Office Hadley Centre Climate
Programme (GA01101). K. Carslaw is currently a Royal Society Wolfson Merit Award holder. We acknowledge funding from NERC under
grants AEROS, ACID-PRUF, GASSP and A-CURE (NE/G006172/1, NE/I020059/1, NE/J024252/1 and NE/P013406/1). This work and its
contributors (J. Johnson, D. Sexton and K. Carslaw) were supported by the UK-China Research & Innovation Partnership Fund through the
Met Office Climate Science for Service Partnership (CSSP) China as part of the Newton Fund. This work used the ARCHER UK National



Supercomputing Service (http://www.archer.ac.uk). ARCHER project allocation n02-FREEPPE and the Leadership Project allocation n02-
CCPPE were used to create the ensemble. The authors appreciate the commitment given by participants in the expert elicitation, particularly
C. Johnson, B. Johnson, J. Mollard, S. Turnock, D. Hamilton, A. Schmidt, C. Scott, R. Stevens, E. Butt, C. Reddington, M. Woodhouse, D.
Spracklen and O. Wild.



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



**Table A1.** Descriptions of the perturbed parameters. Parameters are grouped according to their source within the model as either 'Atm' for atmospheric or 'Aer' for aerosol parameters.

| Name | Source | Description | PDF |
|---|---|---|---|
| Rad_Mcica_Sigma | Atm | Fractional standard deviation of sub-grid condensate seen by radiation | Trapezoid (0.1,0.4,1.5,2.2,2,2) |
| C_R_Correl | Atm | Cloud and rain sub-grid horizontal spatial correlation | Trapezoid (0.0, 0.6, 0.9, 1.0, 1.8, 1. 1, 1.5) |
| Niter_Bs | Atm | Number of microphysics iteration sub-steps | Uniform (5, 20) |
| Ent_Fac_Dp | Atm | Entrainment amplitude scale factor | Trapezoid (0, 0.5, 2, 4, 2, 2) |
| Amdet_Fac | Atm | Mixing detrainment rate scale factor | Trapezoid (0, 0.5, 10.0, 15.0, 2, 2) |
| Dbsdtbs_Turb_0 | Atm | Cloud erosion rate ($s^{-1}$) | Trapezoid (0, 1e-04, 5e-04, 1e-03, 2, 2) |
| Mparwtr | Atm | Maximum value of function controlling convective parcel maximum condensate | Trapezoid (1e-3, 1e-3, 1.5e-3, 2e-3, 2, 2) |
| Dec_Thres_Cld | Atm | Threshold for cloudy boundary layer decoupling | Trapezoid (0.01, 0.011, 0.1, 0.8, 2, 4, 4) |
| Fac_Qsat | Atm | Rate of change in convective parcel maximum condensate | Uniform (0.25, 1) |
| Ageing | Aer | Ageing of hygrophobic aerosols (no. of monolayers of organic material) | Trapezoid (0.3, 1, 5, 10, 2, 2) |
| Cloud_pH | Aer | pH of cloud droplets | Trapezoid (4.6, 5.3, 6.3, 7, 4, 2) |
| Carb_BB_Ems | Aer | Carbonaceous biomass burning emission scale factor | Trapezoid (0.25,0.8,2.2,4,2,2) |
| Carb_BB_Diam | Aer | Carbonaceous biomass burning emission diameter (nm) | Trapezoid (90, 160, 240, 300, 2, 2) |
| Sea_Spray | Aer | Sea spray aerosol emission scale factor | Trapezoid (0.125, 0.6, 3, 8, 4, 3) |
| Anth_SO2 | Aer | Anthropogenic $SO_2$ emission scale factor | Trapezoid (0.6, 0.81, 1.09, 1.5, 2, 2) |
| Volc_SO2 | Aer | Volcanic $SO_2$ emission scale factor | Trapezoid (0.71, 0.99, 1.7, 2.38, 4, 1.1) |
| BVOC_SOA | Aer | Biogenic secondary aerosol formation from volatile organic compounds scale factor | Trapezoid (0.81, 1.08, 3.5, 5.4, 3, 3) |
| DMS | Aer | Dimethylsulphide surface ocean $SO_2$ concentration scale factor | Trapezoid (0.5, 1.26, 1.82, 2, 2, 3) |
| Dry_Dep_Acc | Aer | Accumulation mode dry deposition velocity scale factor | Trapezoid (0.1, 0.32, 3.16, 10, 2, 2) |
| Dry_Dep_SO2 | Aer | $SO_2$ dry deposition velocity scale factor | Trapezoid (0.2, 0.56, 1.78, 5, 2, 2) |
| Kappa_OC | Aer | Köhler coefficient of organic carbon | Trapezoid (0.1, 0.14, 0.25, 0.6, 4, 4) |
| Sig_W | Aer | Updraft vertical velocity standard deviation | Trapezoid (0.1, 0.36, 0.44, 0.7, 2, 2) |
| Dust | Aer | Dust emission scale factor | Trapezoid (0.5, 0.7, 1.4, 2, 2, 2) |
| Rain_Frac | Aer | Fraction of cloud covered area in large-scale clouds where scavenging occurs | Trapezoid (0.3, 0.31, 0.55, 0.7, 2, 3) |
| Cloud_Ice_Thresh | Aer | Threshold of cloud ice fraction above which nucleation scavenging is suppressed | Trapezoid (0.1, 0.105, 0.35, 0.5, 2, 3) |
| BC_RI | Aer | Imaginary part of the black carbon refractive index | Trapezoid (0.2, 0.352, 0.616, 0.8, 4, 2) |
| OC_RI | Aer | Imaginary part of the organic carbon refractive index | Trapezoid (0, 0, 0.05, 0.1, 2, 6) |



**Table A2.** Summary statistics for the pdfs of 1850-2008 aerosol ERF, ERF$_{ARI}$ and ERF$_{ACI}$ presented in Fig. 1. Perturbations to atmospheric and/or aerosol parameters cause the uncertainty in model output in each case. All values are in $\mathrm{W\,m^{-2}}$. For all samples the null hypotheses of equivalent means or standard deviations are rejected at the 99% confidence level using Welch's t (Welch, 1947) and Bartlett (Snedecor and Cochran, 1989) tests respectively.

| Sample | Perturbations | Mean | Standard deviation | 95% Credible interval | Credible range |
|---|---|---|---|---|---|
| ERF | Atmosphere and aerosol | -1.46 | 0.38 | (-2.18, -0.71) | 1.46 |
| | Atmosphere only | -1.51 | 0.25 | (-1.98, -1.04) | 0.94 |
| | Aerosol only | -1.47 | 0.29 | (-2.01, -0.90) | 1.11 |
| ERF$_{ARI}$ | Atmosphere and aerosol | -0.03 | 0.08 | (-0.19, 0.13) | 0.31 |
| | Atmosphere only | 0.00 | 0.04 | (-0.08, 0.08) | 0.16 |
| | Aerosol only | -0.02 | 0.07 | (-0.16, 0.11) | 0.27 |
| ERF$_{ACI}$ | Atmosphere and aerosol | -1.42 | 0.41 | (-2.20, -0.61) | 1.59 |
| | Atmosphere only | -1.51 | 0.29 | (-2.04, -0.96) | 1.08 |
| | Aerosol only | -1.43 | 0.30 | (-1.99, -0.85) | 1.14 |





**Table A3.** Latitude and longitude boundaries for regions R1-R11. Some regional averages are filtered to include only marine or non-marine data.

| Region | Description | Filter | Latitudes | Longitudes |
|--------|-------------|--------|-----------|------------|
| R1 | North Pacific | Marine | 32.5 to 54 | 144 to -125 |
| R2 | East Pacific Stratocumulus Deck | Marine | 16 to 41 | -146 to -104 |
| R3 | Canada | All | 45 to 73 | -115 to -61 |
| R4 | South-east Pacific Stratocumulus deck | Marine | -26 to 1 | -98 to -70 |
| R5 | North Atlantic | Marine | 27 to 59 | -53 to -12 |
| R6 | South-east North Atlantic | Marine | 8.5 to 26 | -44 to -17 |
| R7 | Arctic | Marine | 61 to 89 | -33 to 57 |
| R8 | Europe | All | 37.5 to 71.5 | -12 to 41 |
| R9 | South-east Atlantic Stratocumulus deck | Marine | -18 to 3 | -16 to 13 |
| R10 | North Indian Ocean | Marine | 5.5 to 23 | 63 to 94 |
| R11 | China | Land | 21 to 40 | 98 to 123 |





**Table A4.** Present-day ToA RSR constraints and the resulting 95% credible intervals of 1850 RSR and 1850-2008 aerosol ERF ($\mathrm{W\,m^{-2}}$) for the unconstrained and constrained samples.

| Constraint | Sample size | 2008 RSR | 1850 RSR | 1850-2008 ERF | 1850-2008 ERF credible range |
|---|---|---|---|---|---|
| Unconstrained | 1000000 | (88.9, 120.1) | (87.5, 118.0) | (-2.65, -0.68) | 1.97 |
| CERES (98.3 $\pm$ 0.25 $\mathrm{W\,m^{-2}}$) | 20127 | (98.05, 98.55) | (94.2, 99.3) | (-2.37, -0.59) | 1.78 |
| CERES North Pacific (162.8 $\pm$ 3.3 $\mathrm{W\,m^{-2}}$) | 108493 | (89.6, 106.8) | (88.0, 105.1) | (-2.25, -0.53) | 1.72 |
| Combined constraint | 4699 | (98.1, 98.5) | (95.7, 97.8) | (-2.30, -0.56) | 1.74 |