# Peer review of "Aerosol and physical atmosphere model parameters are both important sources of uncertainty in aerosol ERF"

_Atmospheric Chemistry and Physics, 2018_

## Referee Comment (RC1) · Anonymous Referee #2 · 4 Mar 2018

Review of "Aerosol and physical atmosphere model parameters are both important sources of uncertainty in aerosol" by Leighton Regayre et al.

This manuscript documents a suite of perturbed parameter experiments to try to understand the radiative effects of aerosols, both the direct radiative and indirect cloud effects. The paper builds an 'emulator' based on perturbed parameter experiments, and then uses it to understand the different sensitivities to aerosol and cloud parameters. The paper is topical for ACP, it is well written, and should be publishable with some substantial revisions and clarifications which I outline below.

Most fundamentally, I have some concerns about the quantitative aspects. The abstract

[Figure]

focuses on percentages of variance explained, but this seems to be entirely dependent on the choice of parameters.

In addition, the discussion of a few items needs to be better as noted below. First, the expert elicitation mentioned in passing in the conclusions is interesting, but needs its own section in the results first. As noted below, I'd be interested in a TOA constraint based on minimizing the root mean square spatial difference from observations, not just the mean TOA. In addition, I think perhaps TOA flux should be first section in the manuscript.

Finally I'm not sure how the authors go to regional, that might take a bit more discussion. Is it an entirely different emulator. I think it might be good to show an emulator figure to explain the method a bit more.

Detailed comments are below:

Page 1, L14: This does note seem like much of a change. Also: seems like a large value.

Page 1, L12: is this 60% of the 60% aerosol uncertainty for 4 parameters?

Page 1, L15: isn't the TOA flux 2 orders of magnitude larger than 2wm-2 ERF ($\sim$240 Wm-2)?

Page 6, L181: if the choice of parameters is arbitrary, then the fractions I think in the abstract are arbitrary. Please clarify: I assume some parameters were thrown out, are you sure you have a comprehensive set and another choice of parameters will give the same percentage change?

Page 12, L376: autoconversion is more important but overstated? This is confusing. Please rephrase.

Page 12, L383: is ERF the total? I.e. ari and ACI? Can you separate them?

Page 13, L406: there is an emulator for each gridbox for TOA flux? Maybe a figure is

necessary.

Page 18, L504: how much does this parameter affect TOA flux? Figure 4just shows the ERF. What would a plot of the actual TOA variance look like?

Page 24, L602: this concerns me since I think a different set of parameters will behave differently.

Page 25, L617: how were the regions identified? Sigma? Please clarify.

Page 30, L718: so you have used a total global value. But this can be a result of cancellation of errors. What if you tried to minimize the RMSE v. CERES? Would that give a similar result? Usually the RMSE is better for model evaluation v. Obs.

Page 31, L719:figure 12 is a bit confusing with the multiple axes for ERF over the different periods. Maybe divide it differently?

Page 31, L726: fig 13 is confusing. I take away that the only thing the observations do is constrain rad_mcica_sigma to a narrower range. The discussion is good, but maybe the figure could be simplified.

Page 35, L831: where did you analyze the expert elicitation selection? Did I miss something?

Page 36, L836: how much is that dependent on the parameter set choice?

Page 36, L840: might be good to mention what fraction are natural aerosol emissions or preindustrial background to compare to previous work. How do these results differ from Carslaw et al 2013?

---

## Referee Comment (RC2) · S. J. Ghan (Referee) · 12 Mar 2018

This manuscript is a clear exposition of an important and illuminating study of the contributions of model parameter uncertainty to uncertainty in aerosol radiative forcing.

Major comment

Uncertainty in autoconversion is likely to be major source of uncertainty in both present day radiation and in ERFaci. The manuscript notes the model presently lacks the feedback of aerosol effects on precipitation back onto the aerosols themselves. This feedback can be important in driving the transition from stratiform cloud to pockets of open

celled convection. Moreover, the study does not consider the sensitivity to the representation of autoconversion itself, in particular the dependence of the autoconversion rate on droplet number. I would like to see more discussion of the impacts of these limitations on the conclusions of the study.

Minor comments.

Lines 71-72: uncertainty.

Lines 390-396. The validation of the emulators is never presented. How accurately did the emulators reproduce the simulated fields?

Lines 458-462. Won't the variance for the combined uncertainty be greater if there are no interactions? If independent don't the variances add? Couldn't negatively correlated interactions decrease the uncertainty from the neutral case, and possible produce less uncertainty than with either set of parameters?

Line 704. Make it clear this is global mean. How does this accuracy constraint compare with the accuracy of the emulator?

Lines 714-716. Why focus on the lower bound and not the range?

Lines 756. than the in the. Line 778. equifinality needs to be defined with a reference provided. Section 3.5.4. Explain why this particular region is chose.

Lines 876-877. Please expand on this. Do Johnson et al. explore aerosol optical depth as a constraint?

---

## Author Comment (AC1) · 4 Jun 2018

In our response, referee comments are marked in bold, our responses and original text in plain text, and altered text in the paper in *italic.* Additionally we highlight altered text in the tracked changes pdf as requested.

**Response to reviewer 1 (Steve Ghan)**

**Major comment: Uncertainty in autoconversion is likely to be major source of uncertainty in both present day radiation and in ERFaci. The manuscript notes the model presently lacks the feedback of aerosol effects on precipitation back onto the aerosols themselves. This feedback can be important in driving the transition from stratiform cloud to pockets of open celled convection. Moreover, the study does not consider the sensitivity to the representation of autoconversion itself, in particular the dependence of the autoconversion rate on droplet number. I would like to see more discussion of the impacts of these limitations on the conclusions of the study.**

The following paragraph has been added to the conclusion section immediately before line 864 in the original submission (line 877 in the revised article):

*"One important source of ERF_ACI uncertainty we did not include in our study is the autoconversion rate of cloud drops into rain drops (Michibata et al., 2015, Malavelle et al., 2017 and Toll et al., 2017). Were we to include the autoconversion rate as an additional source of uncertainty the credible range of aerosol ERFs would be larger. If the autoconversion rate were an important cause of uncertainty in both ToA flux and aerosol ERF, the constraint on ERF uncertainty would likely be stronger. However, if autoconversion were to affect ToA flux and aerosol ERF in different ways or to different extents then including this additional source of uncertainty may amplify the equifinality problem by introducing another important degree of freedom. The additional uncertainty from autoconversion could be constrained to a large extent using collocated observations of changes in liquid water path, cloud fraction and aerosol concentrations. We expect such observations of cloud-aerosol relationships will be particularly useful for constraining a model's ability to represent transitions between cloud regimes and we plan to test their efficacy as constraints in the next phase of our research."*

**Minor comment 1: Lines 71-72: uncertainty.**

Fixed.

**Minor comment 2: Lines 390-396. The validation of the emulators is never presented. How accurately did the emulators reproduce the simulated fields?**

Fixed.

Revised text:
"Once emulators were validated, *by ensuring that at least 75% of the validation simulations produced output within the relatively small emulator uncertainty bounds,* new emulators conditioned on output from the 191 perturbed parameter simulations (with better space-filling properties) were created by combining the validation simulations with the original set of simulations."

Minor comment 3: Lines 458-462. Won't the variance for the combined uncertainty be greater if there are no interactions? If independent don't the variances add? Couldn't negatively correlated interactions decrease the uncertainty from the neutral case, and possible produce less uncertainty than with either set of parameters?

Yes, if there were no interactions or compensating effects between the aerosol and atmospheric parameters, the combined uncertainty would be much larger because the uncertainties would be additive. Also, it is feasible that multiple negatively correlated interactions could reduce the uncertainty in either the aerosol-only or atmospheric-only samples when the sets of parameters are combined, as Steve suggests. However, in our case the compensating effects between the groups of parameters are strong enough to prevent the individual uncertainties from being additive, but not strong enough to reduce the uncertainties in either individual sample.

Minor comment 4: Line 704. Make it clear this is global mean. How does this accuracy constraint compare with the accuracy of the emulator?

Fixed.

Revised text:

"Figure 12 shows the effect of constraining the modelled present-day *global, annual mean* RSR to within…"

Revised text:
"We also neglect uncertainty caused by unknown model structural errors (Goldstein et al., 2004, Sexton et al., 2012 and Stier et al., 2013), observation representativeness errors (Schutgens et al., 2017) and the emulators themselves (Oakley and O'Hagan, 2004) *which are of the same order of magnitude as the observational uncertainty.*"

Minor comment 5: Lines 714-716. Why focus on the lower bound and not the range?

We focus on the lower credible interval value because there is substantial debate within the climate forcing community about the validity of the strongest aerosol ERF values and this result contributes to that debate. However, because the credible range is also of interest, we have revised the text.

Revised text:

"The effect of applying the RSR observational constraint is to rule out 1850-2008 aerosol ERF values lower than around -2.4 Wm$^{-2}$, which represents around 15% of the original aerosol ERF range *(-2.7, -0.7 Wm$^{-2}$). However, the 95% credible range is only reduced by around 10% because the distribution of aerosol ERFs in the constrained sample is skewed towards weaker forcings and the upper bound of the credible interval (-0.6 Wm$^{-2}$) is larger (Table A4)."*

Minor comment 6: Lines 756. than the in the. Line 778. equifinality needs to be defined with a reference provided. Section 3.5.4. Explain why this particular region is chose.

Fixed.

Revised text:

"because cloud droplet activation is more sensitive to Sig_W in the present-day atmosphere than *[the]* in the aerosol-limited pre-industrial atmosphere."

We now introduce equifinality, with appropriate citations, when compensating errors are first discussed on line 921 (line 894 of original submission).

Revised text:

"However, despite a very large reduction in plausible parameter space, the effectiveness of the observational constraint is modest because it is hampered by compensating effects between multiple uncertain parameters, *which results in multiple equally plausible solutions (sometimes referred to as `equifinality'; Beven and Freer, 2001, Lee et al., 2016).*"

Revised text:

"Therefore we now use satellite observations of the North Pacific (region R1; latitude 32N-54N; longitude 125W-144E; *the largest regional contribution to global mean aerosol ERF)*".

Minor comment 6: Lines 876-877. Please expand on this. Do Johnson et al. explore aerosol optical depth as a constraint?

Fixed.

Revised text:

"using multiple observable quantities *(e.g. aerosol optical depth and aerosol concentrations)* in Johnson et al. (2018)."

**Response to reviewer 2 (Anonymous reviewer)**

Main Point 1: The abstract focuses on percentages of variance explained, but this seems to be entirely dependent on the choice of parameters.

Indeed. The results from any study depend on the assumptions made. Interested readers will presumably ask the same question, then we trust they will make their own judgement about the completeness of the study by reading the paper. The perturbed parameter ensemble presented here was designed to include perturbations to all processes known to be important for global mean aerosol ERF. The choice of parameters included in the experimental design was informed by expert analyses, including multiple existing perturbed parameter ensembles and our one-at-a-time-screening experiments. We find it difficult to add further detail on the completeness issue in the abstract. However, we have added text to the body of the article.

Additional text:

*"Our results may change slightly with the inclusion of additional parameters. However, we went through a thorough parameter screening and prioritisation process so we consider the parametric uncertainty to be close to an upper limit. Furthermore, with many possible opportunities for parameter compensation, additional parameters only very gradually increase the overall uncertainty."*

Main Point 2: the expert elicitation mentioned in passing in the conclusions is interesting, but needs its own section in the results first.

We describe the use of the expert-elicited pdfs in section 2.3. Furthermore, the expert elicitation process is described in detail elsewhere (e.g. Lee et al, 2012, Yoshioka et al., 2018 and Sexton et al., 2018). The latter two article were not available at the time of review. These articles focus on design aspects of our perturbed parameter ensembles. Therefore, we feel that including a more detailed description of the elicitation process here would duplicate what is presented in these papers (and specifically Yoshioka et al., 2018).

Main Point 3: I'd be interested in a TOA constraint based on minimizing the root mean square spatial difference from observations, not just the mean TOA.

We agree that there are multiple ways to use observations to constrain the aerosol ERF uncertainty and the suggested strategy could provide additional constraint. Constraining the RMSE is akin to constraining the ToA flux in multiple regions. We will follow up on this suggestion in our future research and have revised the text accordingly.

Revised text:

"The aerosol ERF uncertainty decreases when we constrain global mean ToA RSR (-2.37 to -0.59 $Wm^{-2}$) and when we constrain both North Pacific and global RSR (-2.30 to -0.56 $Wm^{-2}$). *These results suggest that additional constraint of aerosol ERF uncertainty could be achieved using multiple regional ToA flux observations."*

Main Point 4: I think perhaps TOA flux should be first section in the manuscript.

We feel that this is a matter of style. Our priority is the reduction of uncertainty in aerosol ERF and its components. Therefore, we quantify the uncertainty and identify the key causes of uncertainty in these terms first. The paper then progresses to an analysis of one observable quantity that may help to constrain the model. Furthermore, the main result of the article is the importance of both aerosol and physical atmosphere parameters as causes of aerosol ERF uncertainty. The existing order of the results section allows us to make these points clearly, whereas the suggested change of order of analyses would obscure what we consider to be the most important results.

Main Point 5: I'm not sure how the authors go to regional, that might take a bit more discussion. Is it an entirely different emulator. I think it might be good to show an emulator figure to explain the method a bit more.

We agree that the transition into regional analyses was abrupt and lacked the important enunciation of the fact that distinct emulators are used for global and regional analyses.

Revised text:

*"Emulators of regional-mean aerosol ERFs and its components were created so that the key causes of variance in each region could be identified (Fig. 11)."*

Specific comment 1: Page 1, L14: This does not seem like much of a change. Also: seems like a large value.

The reviewer is correct that this is a stronger value of forcing than is commonly cited in the literature. However, our values are lower credible bounds on aerosol ERF given our dense sampling of model uncertainty. Such bounded estimates of forcing cannot be achieved using ensembles of well-tuned climate models where individual model uncertainty is neglected (as outlined in Section 1 and in Johnson et al. (2018)). The change in the credible lower bound on aerosol ERF from -2.65 to -2.37 $Wm^{-2}$ is only around 10% of the original lower bound. The change is indeed not substantial, but does serve to illustrate the challenge of reducing uncertainty in complex models. Furthermore, observational constraint of the lower credible bound (regardless of magnitude) adds to the debate within the climate forcing community about the plausibility of the strongest aerosol ERFs.

Specific comment 2: Page 1, L12: is this 60% of the 60% aerosol uncertainty for 4 parameters?

We appreciate the confusion.

Revised text:

*"Four atmospheric and aerosol parameters account for around 80% of the uncertainty in short-wave ToA flux (mostly parameters that directly scale cloud reflectivity, cloud water content or cloud droplet concentrations), and these parameters also account for around 60% of the aerosol ERF uncertainty."*

Specific comment 3: Page 1, L15: isn't the TOA flux 2 orders of magnitude larger than 2wm-2 ERF (~240 Wm-2)?

Fixed.

Revised text:
"the ToA flux is *two orders* of magnitude larger than the aerosol ERF"

"These results counter the belief that observations of ToA reflected short-wave radiation should not constrain the aerosol ERF (because RSR values are *two orders* of magnitude larger than the aerosol ERF)"

Specific comment 4: Page 6, L181: if the choice of parameters is arbitrary, then the fractions I think in the abstract are arbitrary. Please clarify: I assume some parameters were thrown out, are you sure you have a comprehensive set and another choice of parameters will give the same percentage change?

The choice of parameters is not arbitrary. In fact the aim of this paragraph is to explain that the list of parameters was chosen based on expert elicitation – so we think it is as comprehensive as can be achieved based on expert knowledge. Expert elicitation is widely used to make such assessments in many fields. To make this point somewhat clearer, we have included the word 'domain' in the phrase 'model domain experts' and added a sentence at the end of the paragraph: *"Our results may change slightly with the inclusion of additional parameters. However, we went through a thorough parameter screening and prioritisation process so we consider the parametric uncertainty to be close to an upper limit. Furthermore, with many possible opportunities for parameter compensation, additional parameters only very gradually increase the overall uncertainty.".* While we acknowledge that different results would be obtained with a different set of parameters, we consider our set of parameters to sufficiently complete that the results are useful. There is no prior work that enables any estimate of the relative importance of aerosol and atmospheric parameters, so our paper is at least a first step, and the methodology is sufficiently well explained that readers can reach their own judgement.

Revised text:
"model *domain* experts"

Additional text:
*"Our results may change slightly with the inclusion of additional parameters. However, we went through a thorough parameter screening and prioritisation process so we consider the parametric uncertainty to be close to an upper limit. Furthermore, with many possible opportunities for parameter compensation, additional parameters only very gradually increase the overall uncertainty."*

Specific comment 5: Page 12, L376: autoconversion is more important but overstated? This is confusing. Please rephrase.

Fixed. Note that an incorrect reference was used in the initial submission and has now been changed.

Revised text:

"The cloud liquid water path response to aerosols in low, warm clouds is weaker in HadGEM than in other global climate models (Ghan et al., 2016). Therefore, autoconversion may *seem* more important in other models, but will likely be overstated (*Toll et al., 2017*). This process should be considered in future uncertainty analysis studies once shared model structural errors are addressed and the process uncertainty better quantified."

Additional text (Section 4):
*"One important source of ERF_ACI uncertainty we did not include in our study is the autoconversion rate of cloud drops into rain drops (Michibata et al., 2015, Mallavelle et al., 2017 and Toll et al., 2017). Were we to include the autoconversion rate as an additional source of uncertainty the credible range of aerosol ERFs would be larger. If the autoconversion rate were an important cause of uncertainty in both ToA flux and aerosol ERF, the constraint on ERF uncertainty would likely be stronger. However, if autoconversion were to affect ToA flux and aerosol ERF in different ways or to different extents then including this additional source of uncertainty may amplify the equifinality problem by introducing another important degree of freedom. The additional uncertainty from autoconversion could be constrained to a large extent using collocated observations of changes in liquid water path, cloud fraction and aerosol concentrations. We expect such observations of cloud-aerosol relationships will be particularly useful for constraining a model's ability to represent transitions between cloud regimes and we plan to test their efficacy as constraints in the next phase of our research."*

Specific comment 6: Page 12, L383: is ERF the total? I.e. ari and ACI? Can you separate them?

Fixed.

Revised text:
"(*ToA flux, aerosol ERF and its components*)"

Specific comment 7: Page 13, L406: there is an emulator for each gridbox for TOA flux? Maybe a figure is necessary.

We have changed the key sentence to specify that the analyses were applied at all levels for ToA flux as well as forcing terms. We prefer not to add another figure for the ToA flux, but the fraction of uncertainty caused by aerosols and physical atmosphere parameter at the individual gridbox level is shown in Fig 10.

Please note that we have corrected the statement of degraded model resolution.

Revised text:
"Emulation and sensitivity analyses were applied at the individual model gridbox level (degraded to N*48* model resolution) as well as at the regional and global mean level*s for the ToA flux as well as the forcing terms.*"

Specific comment 8: Page 18, L504: how much does this parameter affect TOA flux? Figure 4 just shows the ERF. What would a plot of the actual TOA variance look like?

Fig. 4 shows the causes of variance in aerosol ERF and its components. The equivalent figure showing the fraction of variance in ToA flux is Fig. 7. We have revised the text to better connect the current statements about the aerosol ERF and this parameter to ToA flux which is analysed in section 3.3.

Revised text:
"altering (amongst other things) reflected radiation, tropospheric temperature profiles and cloud amount *(Section 3.3)*."

Specific comment 9: Page 24, L602: this concerns me since I think a different set of parameters will behave differently.

We're unsure what aspect of this sentence concerns the referee. We have included additional text for clarification (Main Comment 1 and Specific Comment 4).

Specific comment 10: Page 25, L617: how were the regions identified? Sigma? Please clarify.

The sentence states the criterion for selecting these regions: "regions of substantial aerosol ERF (stronger than around -2.5 $Wm^{-2}$)". We changed this to be more specific.

Revised text:
"(*ensemble mean* stronger than around -2.5 $Wm^{-2}$)"

Specific comment 11: Page 30, L718: so you have used a total global value. But this can be a result of cancellation of errors. What if you tried to minimize the RMSE v. CERES? Would that give a similar result? Usually the RMSE is better for model evaluation v. Obs.

The RMSE approach suggested by reviewer 2 is equivalent to applying multiple regional observational constraints. We showed that the observed monthly mean regional ToA flux provided little additional constraint on aerosol ERF over and above the global, annual mean ToA flux (Section 3.4). We can expect some modest additional constraint on aerosol ERF uncertainty by extending to more regions, but this is beyond the scope of this paper. We have added text to Section 4.

Additional text:

"*These results suggest that additional constraint of aerosol ERF uncertainty could be achieved using multiple regional ToA flux observations.*"

Specific comment 12: Page 31, L719: figure 12 is a bit confusing with the multiple axes for ERF over the different periods. Maybe divide it differently?

Fixed.

Revised text:
Consequently, the smaller set of model variants also predicts reasonably constrained 1978 and 1850 RSR ranges *(Fig. 12(a))*. However, despite reducing the plausible parameter space by 97% and the RSR range by 98% the impact on the aerosol ERF uncertainty is more modest *(Fig. 12(b) and (c))*.

Revised caption text:
*(a)* Observationally constrained present-day ToA RSR and the values of 1978 and 1850 ToA RSR and *(b)* 1978-2008 and *(c)* 1850-2008 aerosol ERF values from matching model variants.

Specific comment 13: Page 31, L726: fig 13 is confusing. I take away that the only thing the observations do is constrain rad_mcica_sigma to a narrower range. The discussion is good, but maybe the figure could be simplified.

Yes, the Rad_Mcica_Sigma parameter is constrained. The remainder of Section 3.5.2 contains important insights regarding why this constraint does not feed through to aerosol ERF, when that parameter dominates the uncertainty. This discussion informs the reader why we need to explore the constraint using parameter relationships. Although it's quite complicated, the figure content is all relevant to the discussion in this section.

Specific comment 14: Page 35, L831: where did you analyze the expert elicitation selection? Did I miss something?

This was presented on lines (436 – 439 of original submission; 440-443 of revised article) in section 3.1 where we quantify the effect of the expert elicited pdfs on the aerosol ERF uncertainty. We have cross-referenced Section 3.1 within Section 4, to ensure readers do not overlook this important result.

Revised text:
"can be achieved by applying probability distributions to the parameters based on expert elicitation *(Section 3.1).*"

Specific comment 15: Page 36, L836: how much is that dependent on the parameter set choice?

Fixed.

Additional text:
*"Our results may change slightly with the inclusion of additional parameters. However, we went through a thorough parameter screening and prioritisation process so we consider the parametric uncertainty to be close to an upper limit. Furthermore, with many possible opportunities for parameter compensation, additional parameters only very gradually increase the overall uncertainty."*

Specific comment 16: Page 36, L840: might be good to mention what fraction are natural aerosol emissions or preindustrial background to compare to previous work. How do these results differ from Carslaw et al 2013?

Fixed.

Revised text (Section 3.2.1):
"Natural aerosol emissions (here, *predominantly* Sea_Spray, DMS and BB_Diam) persist as important sources of industrial-period $ERF_{ACI}$ uncertainty, as in previous studies of several climate models (Wilcox et al., 2015) and the aerosol-only component of a global model (Carslaw et al., 2013). *Here, natural aerosols are responsible for around 63% of the proportion of $ERF_{ACI}$ variance caused by aerosol parameters, compared to 45% of the variance in aerosol-cloud-albedo effect forcing in the absence of rapid atmospheric adjustments (Carslaw et al., 2013).* However, by far the largest source of uncertainty is the Rad_Mcica_Sigma parameter."

Revised text (Section 4, although not exactly where suggested by reviewer 2):
"Therefore, the causes of aerosol ERF uncertainty in recent decades (1978-2008) are model deposition rates (model process parameters) and anthropogenic emissions, whilst the 1850-2008 aerosol ERF is most sensitive to natural aerosol emissions *(which collectively cause around 63% of the aerosol contribution to $ERF_{ACI}$ variance).* The magnitude of global mean aerosol forcing on the decadal timescale depends on the combination of uncertain positive and negative regional forcings (Regayre et al., 2015; Fig. 5)."